# TASK-ALIGNED ATTENTION RETRIEVAL FOR SCALING TABULAR FOUNDATION MODELS

## ABSTRACT

Retrieval serves as an effective approach to address various scaling challenges in in-context learning with tabular foundation models, yet prevailing methods select neighbors by Euclidean proximity in the covariates and thus ignore how the task mapping varies across the feature space. We introduce Task-Aligned Attention Retrieval (TAAR), a simple, model-agnostic procedure that, for each query, selects the most predictive features and relevant context samples using the model's own attention scores. TAAR therefore ranks candidates by a task-aligned similarity already internalized by the foundation model, rather than by raw geometric distance in input features. TAAR is a drop-in module for state-of-the-art tabular foundation models (e.g., TabPFN and LimiX), requires no fine-tuning, and adds only an extra forward pass. On classification and regression benchmarks, TAAR achieves pronounced gains in accuracy and stability over current retrieval methods and supports scaling along feature space, sample size and target-class cardinality.

## 1 INTRODUCTION

Foundation models for tabular data are rapidly maturing, with transformer-based architectures and PFN-style pretraining showing compelling performance, especially in small-to-medium data regimes where classical gradient-boosted trees long dominated (Hollmann et al., 2023; 2025; Prokhorenkova et al., 2018; Huang et al., 2020; Gorishniy et al., 2021; Arik & Pfister, 2021; Somepalli et al., 2021). Recent tabular foundation models, including most notably TabPFN and follow-ups, leverage large-scale synthetic priors and in-context learning (ICL) (Hollmann et al., 2023; Nagler et al., 2023; Hollmann et al., 2025; Jingang et al.). This is precisely how emerging tabular foundation models operate in practice, amortizing Bayesian or kernel-like inference across massive synthetic task corpora to deliver strong zero-/few-shot accuracy and latency gains on small- to medium-sized datasets.

However, several persistent bottlenecks motivate explicit retrieval for scaling these foundation models at test time. First, tabular tasks exhibit long-tail categoricals, mixed types, and covariate shift across sources Van Breugel & Van Der Schaar (2024), so excessive context length or high contextual complexity (e.g., heterogeneous or conflicting demonstrations) dilutes informative evidence and induces interference, which in turn degrades predictive performance (Thomas et al., 2024a). Second, because memory scales quadratically with context size, performance is curtailed when the dataset cannot fit in memory, unlike classical methods whose accuracy typically increases with additional data. Third, existing tabular foundation models are typically trained only on classification tasks with a limited number of classes (usually up to 10), making them unsuitable for direct application to tasks exceeding 10 classes. Current adaptation methods often need to employ a divide-and-conquer strategy with multiple inference passes (Ye et al., 2024).

While recent retrieval strategies for tabular foundation models scale to large corpora, they exhibit a fundamental limitation: exemplar selection is typically driven by proximity in the input feature space, rather than by the predictive mapping from features to targets, so they cannot support feature-level and category-level scaling

(Thomas et al., 2024a; Xu et al., 2024). In practice, this input-distance-only criterion is fragile in mixed-type, high-dimensional tables, where distance concentration and hubness make "nearest" neighbors unreliable and repeatedly surface misleading hubs (Beyer et al., 1999; Radovanović et al., 2010). Decades of metric-learning theory show that task-aligned (label-aware) similarities—learned from input-target pairs consistently reduce k-NN error relative to Euclidean distance, by enlarging class margins and aligning neighborhoods with decision boundaries (Goldberger et al., 2005; Weinberger & Saul, 2009; Xing et al., 2002). Contemporary tabular ICL pipelines that retrieve nearest neighbors (e.g., LoCalPFN (Thomas et al., 2024b)) therefore inherit the weaknesses of raw feature-space geometry.

To address these issues, we propose Task-Aligned Attention Retrieval (TAAR), which ranks candidates using the model's own attention scores—an implicit, kernel-like similarity trained to be predictive—thereby coupling retrieval to the task function and overcoming the core bottleneck of geometry-only selection (Choromanski et al., 2021; Katharopoulos et al., 2020). In tabular foundation models such as TabPFN v2 and LimiX, attention learned from context examples often concentrates on variables mediating the predictive signal; under standard structural assumptions (Markovian structure, faithfulness, and no unobserved confounding among selected parents), the feature-attention distribution prioritizes direct causal parents (Zhang et al., 2025), improving robustness to spurious correlations and covariate shifts and stabilizing predictions across scales.

To scale along feature space and sample size, TAAR employs a dual-attention design: a feature-attention module yields a sparse, query-specific weighting over columns to highlight variables most predictive for the current input, while a cross-instance attention module ranks and selects exemplars with the highest task-aligned relevance. This produces contexts that are simultaneously feature-focused and example-focused, reducing computational burden and enabling scalable selection at both the feature and sample levels. To scale along the target category number without altering the backbone, we further introduce Class-Range Lifting Retrieval (CRLR), a regression-guided application of TAAR that constructs a label-sparse local context so the effective number of classes seen at inference is far smaller than the global cardinality, preserving latency while improving accuracy and stability. We provide a rigorous theoretical analysis of TAAR and validate its efficacy with comprehensive experiments on various tabular benchmarks. Empirically, TAAR achieves favorable scaling properties across feature space, sample size, and target-class cardinality. It generally outperforms state-of-the-art baselines, achieving a competitive balance between accuracy and efficiency.

## 2 METHOD

We present a scalable, task-aligned methodology for tabular ICL that converts three traditional bottlenecks into levers for accuracy and efficiency: (i) many features (high-dimensional, mixed-type inputs), (ii) many context samples (long prompts), and (iii) many target classes (high-cardinality outputs). Our approach uses a dual-attention principle, feature-wise attention to focus the representation on informative variables and instance-wise attention to retrieve prediction-relevant exemplars, together with a class-range lifting mechanism that enables large-class classification without modifying the backbone model.

**Notations.** We consider a multi-class classification problem with $K$ classes. Let $(X, Y) \sim \mathsf{P}$ denote a random pair, where $X \in \mathbb{R}^d$ is the feature vector and $Y \in [K] = 1, \ldots, K$ is the class label. A tabular foundation model $f$ takes as input a query $X_{\text{test}} \in \mathbb{R}^d$ together with a context $C = (X_i, Y_i)_{i=1}^{L}$, where $(X_i, Y_i)_{i=1}^{L}$ are i.i.d. samples drawn from $\mathsf{P}$. The model outputs a prediction $\hat{Y} = h_\theta(X_{\text{test}} \mid C)$. We denote by $\text{Attn}^{\text{feat}}$ the feature-level attention across columns, and by $\text{Attn}^{\text{inst}}$ the instance-level attention from the query token to the context tokens.

## 2.1 FEATURE ATTENTION GUIDED SUBSPACE SELECTION

High-dimensional tables contain irrelevant or redundant variables that inflate computation and obscure signal. Yet current tabular ICL retrieval methods only consider sample retrieval instead of feature retrieval (Thomas et al., 2024a; Xu et al., 2024). We instead read out the model's own attention, which is already trained on $(X, Y)$ to produce per-query feature importance and construct a compact, information-dense view of the table.

For query $X$ and candidate context $C$, we aggregate feature-level attention to score column $j \in [d]$:

$$a_j = \underset{h \in \mathcal{H}}{\mathrm{Agg}} \Big( \underset{t \in [d] \setminus \{j\}}{\mathrm{Pool}} \, \mathrm{Attn}^{\mathrm{feat}}_{j \to t, \, h} \Big), \tag{1}$$

where $\mathcal{H}$ indexes heads, `Pool` averages or max-pools over tokens, and `Agg` averages over heads (optionally with temperature scaling). For prediction tasks such as classification and regression, we only consider the attention score to the target column. We determine the feature budget $m$ as the minimum of two criteria, balancing attention coverage and a global proportion cap. Let $a_{(1)} \geq \cdots \geq a_{(d)}$ be the scores in descending order and $A = \sum_{j=1}^{d} a_j$. Define the attention-coverage budget

$$m_{\mathrm{att}} = \min \Big\{ m \in \{1, \ldots, d\} : \sum_{j=1}^{m} a_{(j)} \geq \rho_{\mathrm{att}} S \Big\},$$

with coverage threshold $\rho_{\mathrm{att}} \in (0, 1]$ (e.g., 0.85–0.95), and the proportional cap

$$m_{\mathrm{prop}} = \lceil p_{\max} d \rceil,$$

with $p_{\max} \in (0, 1]$ (e.g., 0.1–0.3). We then set

$$m = \min\{ m_{\mathrm{att}}, \, m_{\mathrm{prop}} \},$$

optionally clamped to $[m_{\min}, m_{\max}]$ for stability across queries. We obtain a *feature mask* $M \subseteq [d]$ via *Top-m* selection $M = \mathrm{topm}(a)$. Let $\mathbf{b} \in \{0, 1\}^d$ be the indicator of $M$ with $b_j = \mathbb{I}[j \in M]$; we form masked inputs

$$\tilde{X} = X \odot \mathbf{b}, \qquad \tilde{C} = \{(\tilde{X}_i, Y_i)\},$$

and use $(\tilde{X}, \tilde{C})$ for the following prediction.

Because attention is learned from $(X, Y)$, $s_j$ emphasizes variables that are predictive for the current query, not merely globally variant. Under sparse-parent SCM assumptions (faithfulness, no hidden confounding among selected parents), feature attention concentrates on the Markov blanket, reducing spurious associations and improving shift robustness. Please refer to Sec. 3.1 for empirical evidence. Replacing $d$ with $m \ll d$ lowers projection FLOPs and memory; any column-wise attention also scales in $m$. The theoretical analysis of feature selection is as follows.

**Theoretical analysis.** We now show that selecting a subset of features can possibly improve the generalization error of the resulting model. Specifically, we assume that only a subset of features is truly relevant for prediction. Regarding ICL, although Han et al. (2025) demonstrate that it can be well approximated by kernel regression, we adopt $k$-NN classification as a tractable surrogate for the inference stage in our analysis.

In detail, we first assume that there exists a measurable mapping $g^\star : \mathbb{R}^d \to \mathbb{R}^m$ ($m \ll d$), defining the sufficient feature $S = g^\star(X) \in \mathbb{R}^m$, satisfying the *sufficiency condition* $Y \perp\!\!\!\perp X \mid S$. And S is learnable to a foundation model, as we show empirically in Sec. 3.1. Thus, the Bayes optimal classifier depends only on $S$:

$$\eta_k^\star(s) = \mathsf{P}(Y = k \mid S = s), \qquad f^\star(x) = \arg\max_{k \in [K]} \eta_k^\star\big(g^\star(x)\big).$$

Given i.i.d. training samples $\{(S_i, Y_i)\}_{i=1}^n$ (where $S_i = g^\star(X_i)$), for any point $s \in \mathcal{S}$, let $\mathcal{N}_k(s)$ denote the $k$ nearest neighbors of $s$ (Euclidean metric). Then the $k$-NN classifier $\hat{f}_{k\text{-NN}}$ is defined as follows.

$$\widehat{\eta}_k(s) = \frac{1}{k} \sum_{i \in \mathcal{N}_k(s)} \mathbf{e}(Y_i) \in \Delta^{K-1}, \qquad \hat{f}_{k\text{-NN}}(s) = \arg \max_k \left[\widehat{\eta}_k(s)\right]_k.$$

We could now get the generalization error of the obtained classifiers.

**Theorem 2.1** (Informal; see Theorem B.5). *Under certain assumptions, by selecting an appropriate $k$, we obtain*

$$\mathcal{R}(\hat{f}_{k\text{-NN}}) - \mathcal{R}(f^\star) \leq \tilde{O}\left(n^{-\frac{\beta(1+\alpha)}{2\beta+m}}\right). \tag{2}$$

*Here, $\mathcal{R}(f) := \mathsf{P}(f(S) \neq Y)$ denotes the classification error rate, and $\alpha, \beta$ are constants independent of $n$.*

**Corollary 2.2.** *If $k$-NN is applied directly to $X \in \mathbb{R}^d$ under the same conditions as in Theorem 2.1, and the resulting classifier is denoted by $\hat{f}_{k\text{-NN}}^{\text{full}}$, then by selecting an appropriate $k$, we obtain*

$$\mathcal{R}(\hat{f}_{k\text{-NN}}^{\text{full}}) - \mathcal{R}(f^\star) \leq \tilde{O}\left(n^{-\frac{\beta(1+\alpha)}{2\beta+d}}\right). \tag{3}$$

*Here, $\alpha$ and $\beta$ are the same constants as those in Theorem 2.1.*

Note that when $m < d$, the right-hand side of Eq. (2) is strictly smaller than that of Eq. (3), implying that $k$-NN may achieve improved generalization performance with reduced data.

## 2.2 CROSS-ATTENTION BASED INSTANCE SELECTION

ICL quality hinges on which exemplars enter the prompt. Current distance-only retrieval in raw/embedded $X$ returns neighbors that may be geometrically close yet predictively uninformative and yields long, interference-prone context. We rank candidates by the query's instance attention, which serves as a task-aligned similarity.

We compute query-to-instance cross-attention on masked features

$$\gamma_i = \mathrm{Attn}_{\psi_t(Y_{\text{test}}) \to \psi_c(Y_i)}^{\text{inst}}, \qquad i \in [L], \tag{4}$$

where $\psi_t(Y_{\text{test}})$ and $\psi_c(Y_i)$ are the label embeddings for test and context samples. Then we select the top-$k$ exemplars $C_k(x) = \mathrm{topk}(\alpha)$. We determine the select ratio using the same criteria in Sec. 2.1 with different thresholds. Implementation details are shown in Sec. C.

Replacing $L$ with $k \ll L$ shrinks attention maps from $O(L^2)$ to $O(k^2)$ and reduces token-wise compute, supporting scaling contexts without losing predictive information.

For datasets with an extremely large number of cells, computing query-context attention over the entire pool is infeasible under given GPU memory budgets. We therefore adopt a stratified, streaming instance-selection procedure. Concretely, we partition the context set into groups $\{\mathcal{B}_g\}_{g=1}^G$ sized to fit the memory constraint, with each group formed by stratified sampling so that its target distribution closely matches that of the full pool. We then process the groups sequentially: for each $\mathcal{B}_g$, we compute attention-based relevance scores for all candidates, retain the top-ranked subset, and merge them into a global candidate set that is carried forward to the next group. Iterating over all groups yields a final shortlist comprising the globally most predictive instances, while operating at $\mathcal{O}(B_{\max}^2)$ memory per step rather than $\mathcal{O}(L^2)$ for the full pool. This streaming design progressively accumulates high-utility exemplars and preserves task alignment, enabling attention-based retrieval at scales that would otherwise exceed hardware limits.

**Theoretical analysis.** We define the kernel on $S$-space as $K_S(s, s') := \kappa\big(\|\phi(s) - \phi(s')\|\big)$, where the map $\phi : \mathcal{S} \to \mathbb{R}^m$ is bi-Lipschitz, and the kernel scalar function $\kappa : [0, \infty) \to (0, \infty)$ is a radial, monotonically decreasing, Lipschitz, bounded kernel scalar function. We consider two truncated Nadaraya-Watson estimators: Method 1 selects $k$ nearest neighbors in $X$-space using Euclidean metric, Method 2 selects top $k$ kernel values in $S$-space. It can be proved that under certain assumptions, the guaranteed bound of mean squared error (MSE) of Method 2 is strictly better than Method 1 when $m < d$. Specifically,

**Theorem 2.3.** *Let $x_{test} = (s_{test}, z_{test})$ be the query point. Let $J_k(x_{test}) \subseteq \{1, \ldots, n\}$ be the indices of the $k$ smallest Euclidean distances $\|X_i - x_{test}\|$. Define the $X$-space estimator as*

$$\widehat{\eta}^{(1)}(s_{test}) := \frac{\sum_{i \in J_k(x_{test})} K_S(s_{test}, S_i)\, \mathbf{e}(Y_i)}{\sum_{i \in J_k(x_{test})} K_S(s_{test}, S_i)};$$

*Let $L_k(s_{test})$ be the index set of the top $k$ largest $K_S(s_{test}, S_i)$ (equivalent to the $k$ smallest $\|\phi(S_i) - \phi(s_{test})\|$). Define the $S$-space estimator as*

$$\widehat{\eta}^{(2)}(s_{test}) := \frac{\sum_{i \in L_k(s_{test})} K_S(s_{test}, S_i)\, \mathbf{e}(Y_i)}{\sum_{i \in L_k(s_{test})} K_S(s_{test}, S_i)}.$$

*Then under certain assumptions, we have*

$$\mathrm{MSE}_1(s_{test}) := \mathbb{E}\big[\|\widehat{\eta}^{(1)}(s_{test}) - \eta(s_{test})\|_2^2\big] \le \Theta\big(n^{-\frac{2\beta}{2\beta+d}}\big),$$

$$\mathrm{MSE}_2(s_{test}) := \mathbb{E}\big[\|\widehat{\eta}^{(2)}(s_{test}) - \eta(s_{test})\|_2^2\big] \le \Theta\big(n^{-\frac{2\beta}{2\beta+m}}\big)$$

### 2.3 REGRESSION-GUIDED CONTEXT PRUNING FOR CLASS-RANGE LIFTING

Pretrained tabular foundation models for classification are typically constrained by a fixed label vocabulary (e.g., $K_{\max} \le 10$) and cannot natively encode categories beyond this preset range. To circumvent this limitation during retrieval, we employ the regression variant of the backbone to drive both feature and instance selection. Most large tabular models provide regression capability (e.g., TabPFN v2 includes a dedicated regression model; LimiX supports a regression head), allowing us to recast multi-class labels into a continuous target space and compute cross-example attention through the following pipeline.

1. **Regression-guided selection.** We encode class labels into a regression-compatible representation and use a regression head of TFMs (such as LimiX) or TFMs for regression (such as TabPFN v2 regressor) to run FSA and TAAR-$k$. Because the retrieval is label-aware, the selected neighborhood becomes label-sparse: the number of distinct classes among the retrieved context samples typically collapses (often $\le 10$) after a single iteration.

2. **Class-compatible classification.** Let $U_k(X)$ be the set of unique labels in $C_k(X)$. If $|U_k| \le K_{\max}$, we pass $(X, \tilde{C}_k)$ to the classification model. Otherwise, we refine once (re-retrieve, or keep the top-$K_{\max}$ categories by frequency) to meet capacity.

We show details in practice in Sec. C.

## 3 EXPERIMENTS

In this section, we evaluate the performance of TAAR across various tasks. All experiments are performed using an NVIDIA H20 Tensor Core GPU with 96GB memory.

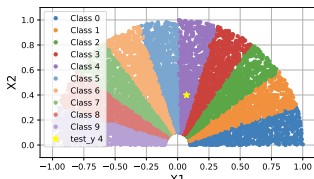
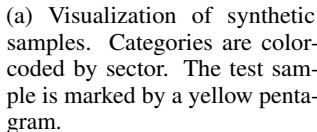
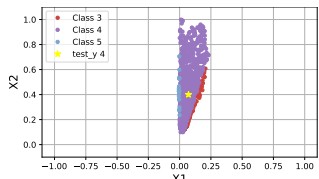
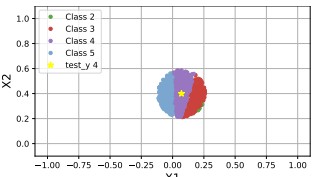

(a) Visualization of synthetic samples. Categories are color-coded by sector. The test sample is marked by a yellow pentagram.

(b) Top 10% of context samples sorted by sample-level attention scores.

(c) Top 10% of context samples sorted by Euclidean distance of input features.

Figure 1: Visualization of retrieval based on sample-level attention. TAAR assigns higher scores to in-context samples from the same category as the query to help the prediction while the retrieval based on Eulidean distance of raw input features retrieves more unrelated samples.

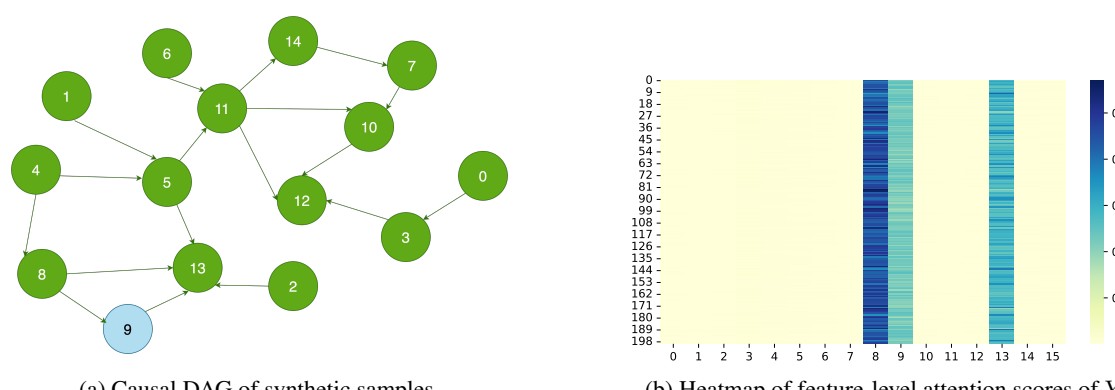

(a) Causal DAG of synthetic samples.

(b) Heatmap of feature-level attention scores of $Y$.

Figure 2: Toy example of feature-level attention for the outcome variable. Direct causes of the outcome and the outcome itself are assigned most of the attention weights in LimiX.

## 3.1 RETRIEVAL VISUALIZATION AND ANALYSIS

To understand the ability of attention layers of current tabular foundation models, we conduct the following experiments and analysis. We employ LimiX as the base model to conduct the following experiments.

**Feature-Attention Visualization** We study how features with different structural–causal relationships manifest inside the attention layers. To this end, we generate synthetic datasets from randomly sampled structural causal models (SCMs). Because each SCM specifies the ground-truth generative dependencies among features, we can directly compare these dependencies with the model's attention weights to assess whether the model has learned to prioritize, for a given target variable, the features with the strongest predictive influence rather than spurious correlates. The synthetic dataset is generated according to the SCM in Fig. 2a, with observed features (green) and the outcome (yellow). The model is constructed such that each causal edge is parameterized by a two-layer ReLU MLP plus Gaussian noise. Results in Fig. 2b reveal that the attention module assigns the highest weights to the precise set of features that are direct causes of the outcome in the SCM. This provides evidence that feature-level attention helps isolate causal drivers and may reduce sensitivity to non-causal, spurious associations.

**Sample-Attention Visualization**   We evaluate whether the attention mechanism in tabular foundation models can identify the most predictive samples. Specifically, we design controlled simulations that analyze the distribution of samples selected by attention scores and compare it against selections based solely on Euclidean distance in the input feature space. This comparison reveals whether attention prioritizes samples that are informative for prediction rather than merely proximal in raw feature space.

A 2D synthetic dataset of 10 classes is generated, with each class distributed within a specific circular sector (shown in Fig. 1a). Given that the attention mechanism computes sample similarity in a latent embedding space—while also accounting for dependencies between input features and the class label—it can model more complex relationships than a simple Euclidean distance applied directly to the original 2D features. A key observation from Fig. 1b is that the attention weights are predominantly assigned to in-context samples of the same class as the query. This indicates that the model effectively leverages class-consistent information by focusing on the most relevant contextual examples.

## 3.2   REAL-WORLD EVALUATIONS

We validate the effectiveness of TAAR in three real-world scenarios: (1) integrating TAAR with prior state-of-the-art tabular foundation models and evaluating on standard benchmarks; (2) comparing against strong retrieval baselines on datasets with massive sample sizes; and (3) assessing scaling behavior on large–label-space classification tasks.

### 3.2.1   EVALUATION ON STANDARD BENCHMARKS

To evaluate TAAR on standard benchmarks, we conduct experiments on both classification and regression benchmarks. The classification benchmarks include BCCO-CLS Zhang et al. (2025), Tabzilla McElfresh et al. (2023), while the regression benchmarks comprise Talent Liu et al. (2024). Performance is compared against state-of-the-art models and KNN-based methods Xu et al. (2024); Thomas et al. (2024a). To accommodate all baseline methods, we set the following computational limits across all tasks: a maximum of 10,000 features, 50,000 samples, and (for classification) 10 classes and max cell to 5,000,000. Detailed descriptions of the settings and datasets are provided in the Sec. C. Experimental results on the Tabzilla Tab. 1, BCCO-CLS Sec. C and Talent Tab. 2 benchmarks show that TAAR brings significant benefits: both LimiX and TabPFN exhibit improved mean performance and ranking on all metrics when augmented with our method. In particular, LimiX achieves state-of-the-art performance when integrated with TAAR.

Table 1: Performance on Tabzilla benchmark. Red and green values indicate performance improvement and degradation relative to the baseline method, respectively.

| Method | AUC (↑) | Acc.(↑) | f1 (↑) | AUC-Rank (↓) | Acc.-Rank (↓) | f1-Rank (↓) |
|---|---|---|---|---|---|---|
| LimiX | 0.934 | 0.855 | 0.795 | 6.333 | 7.074 | 7.444 |
| +KNN | 0.940(+0.006) | 0.881(+0.026) | 0.827(+0.032) | 4.519(-1.814) | 4.407(-2.667) | 4.481(-2.963) |
| +TAAR(ours) | **0.951**(+0.017) | **0.904**(+0.049) | **0.864**(+0.069) | **1.519**(-4.814) | **1.444**(-5.630) | **1.444**(-6.000) |
| TabPFN-v2 | 0.924 | 0.857 | 0.789 | 7.185 | 6.815 | 6.852 |
| +KNN | 0.914(-0.010) | **0.871**(+0.014) | 0.797(+0.008) | 7.519(+0.334) | 6.074(-0.741) | 6.407(-0.445) |
| +TAAR(ours) | **0.934**(+0.010) | 0.870(+0.013) | **0.800**(+0.011) | **4.778**(-2.407) | **5.074**(-1.741) | **6.000**(-0.852) |
| TabICL | 0.929 | 0.851 | 0.798 | 6.963 | 6.444 | 6.556 |
| Mitra | 0.919 | 0.847 | 0.774 | 9.037 | 7.963 | 8.333 |
| XGBoost | 0.929 | 0.863 | 0.789 | 6.778 | 7.000 | 7.333 |
| CatBoost | 0.922 | 0.848 | 0.780 | 8.593 | 8.370 | 8.333 |
| LightGBM | 0.927 | 0.863 | 0.796 | 7.185 | 6.630 | 6.852 |
| TabR | 0.904 | 0.851 | 0.789 | 9.148 | 8.444 | 7.963 |

### 3.2.2 EVALUATION ON DATASETS WITH MASSIVE SAMPLE SIZE

To address the performance degradation and out-of-memory (OOM) issues encountered by models on large-scale datasets, we propose a multi-stage retrieval strategy. For evaluation, we selected 10 large-scale datasets with context lengths exceeding 74,498(up to 1,048,575) from Kaggle, and compared our method against several baselines, including the random sampling method and KNN-based method Xu et al. (2024); Thomas et al. (2024a). The training samples are initially partitioned into $n$ random batches. Retrieval is performed for each test sample in an iterative fashion: at each step, $k$ points retrieved from the current batch are concatenated to the next batch before the subsequent retrieval operation. The cumulative set of points retrieved after processing all batches constitutes the context for inference. The batch size is determined by a trade-off between computational efficiency and memory constraints, with the maximum feasible size selected for each dataset without exceeding GPU memory limits. To manage inference time, we imposed an upper limit of 10,000 test samples for each dataset. We employed the train_test_split function from scikit-learn library, ensuring that the stratification of the target variable y is preserved across the splits. Detailed descriptions of the datasets are provided in Tab. 12. The results of the experiment is shown at Sec. C.3

### 3.2.3 EVALUATION ON LARGE-LABEL-SPACE DATASETS

Additionally, since pre-trained models such as TabPFN and LimiX fix the maximum number of classes in their classification decoder to 10 during pre-training, the upper limit for the number of classes in classification tasks during inference is constrained to 10. Several works Hollmann et al. (2025) have proposed Error-Correcting-Output-Codes(ECOC) Dietterich & Bakiri (1994) on pre-trained model. However, this approach has demonstrated limited effectiveness in practice. Our evaluation focused on multi-class classification tasks from the Talent benchmark Liu et al. (2024). Using accuracy as the evaluation metric, we selecte 12 datasets with more than 10 classes as the evaluation benchmark. Within this framework, the proposed TAAR method is compared against an Error-Correcting Output Codes (ECOC) approach, which shares LimiX as the same base model, as well as other relevant tree-based models. The details of the dataset are provided in the Tab. 11.

The results shown in Fig. 4 highlight the strong performance of tabular foundation models in multi-class classification. TAAR significantly outperforms existing methods on the benchmark, attaining a top accuracy of 87.2%. It is clear that TAAR successfully transcends the limitation of label scaling of ICL-based models. More importantly, it surpasses alternative approaches like tree-based models and the ECOC framework, establishing its superior capability on tasks with more than 10 categories.

## 3.3 SCALING ANALYSE

In this section, we evaluate how TAAR and KNN baseline perform with respect to scaling along three dimensions: sample size, feature count, and number of classes. To this end, we grouped the datasets from the classification benchmark into buckets according to each dimension and compared the relative improvement of both methods over the baseline. The detailed results are presented in the Fig. 3.

## 3.4 ABLATION STUDY

In this section, we provide ablation studies based on LimiX under Tabzilla benchmarks to assess the individual contribution of each component in TAAR. We compared the full model against: (1) a variant without sample attention, (2) a variant without feature attention, and (3) a variant without the dynamic thresholding mechanism, along with baseline methods. The result is shown at Fig. 5

Table 2: Performance Comparison of Tabular Data Modeling Methods On Talent regression task. Red values indicate performance improvement relative to the baseline method, while green values indicate degradation.

| Method | R2(↑) | R2-Rank (↓) |
|---|---|---|
| LimiX | 0.715 | 4.333 |
| +KNN | 0.718(+0.003) | 3.982(-0.351) |
| +TAAR(ours) | **0.737**(+0.022) | **2.323**(-2.010) |
| TabPFN-v2 | 0.686 | 4.869 |
| +KNN | 0.694(+0.008) | 4.834(-0.035) |
| +TAAR(ours) | **0.718**(+0.032) | **4.535**(-0.334) |
| XGBoost | 0.710 | 5.293 |
| LightGBM | 0.707 | 5.596 |
| CatBoost | 0.700 | 6.788 |
| ModernNCA | 0.648 | 8.485 |

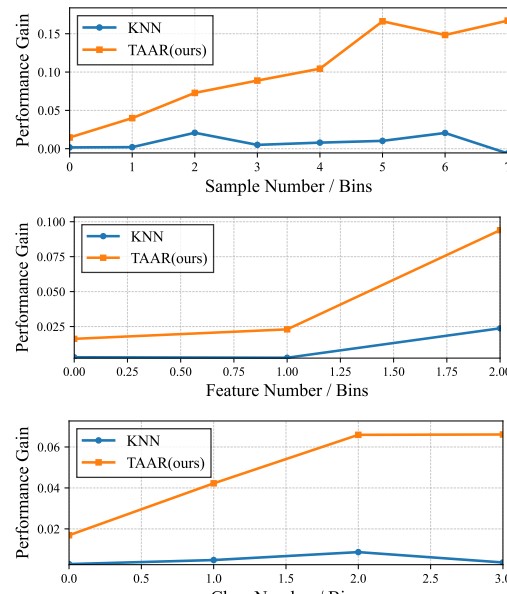

Figure 3: Performance gain of TAAR and KNN baseline across three scaling dimensions.

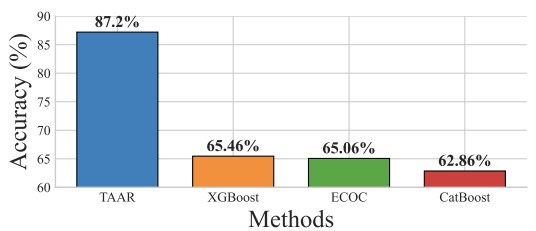

Figure 4: Performance comparison of various methods on multi-class classification problems, measured by Accuracy.

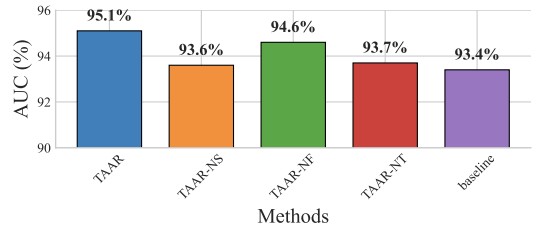

Figure 5: Ablation result.
NS represents TAAR without sample level retrieval.
NF represents TAAR without feature level retrieval.
NT represents TAAR without threshold.

## 4    CONCLUSION

We presented TAAR, a unified framework that scales tabular foundation models along feature space, context size, and target-class cardinality. Experiments confirm consistent improvements over strong baselines.

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

## A   RELATED WORK

**Tabular foundation models and deep tabular baselines.**   Transformer-based architectures and PFN-style priors have advanced the state of the art in tabular learning, delivering strong zero-/few-shot performance via in-context inference (Hollmann et al., 2023; Nagler et al., 2023; Hollmann et al., 2025; Jingang et al.). These approaches complement (and often surpass) classical gradient-boosted trees and modern deep tabular baselines, including XGBoost, LightGBM, CatBoost, TabTransformer, FT-Transformer, SAINT, and TabNet (Chen & Guestrin, 2016; Ke et al., 2017; Prokhorenkova et al., 2018; Huang et al., 2020; Gorishniy et al., 2021; Somepalli et al., 2021; Arik & Pfister, 2021). Our work builds on this line by focusing on test-time retrieval for in-context learning (ICL) with foundation models.

**ICL theory and attention as a kernel similarity.**   A growing body of theory views ICL in transformers as implementing classical estimators (e.g., ridge regression, gradient descent, Bayesian inference) within the forward pass, implying that the similarity used by attention is task-aligned (Akyürek et al., 2022; von Oswald et al., 2023; Xie et al., 2022). This perspective dovetails with work that makes attention's kernel structure explicit (e.g., linear and Performer attention), motivating the use of attention scores as a learned similarity for retrieval (Katharopoulos et al., 2020; Choromanski et al., 2021). TAAR operationalizes this idea for tabular FMs by ranking features and instances via the model's own attention.

**Retrieval for tabular ICL and its limitations.**   Recent efforts adapt retrieval to tabular FMs, e.g., LoCalPFN and related methods that pair retrieval with in-context predictors (Thomas et al., 2024b;a; Xu et al., 2024). However, most existing strategies rely on feature-space distance or unsupervised embedding similarity, which can fail to capture the input-to-target mapping and thus struggle to scale at the feature and category levels. TAAR departs from distance-only heuristics by using label/task-aware attention for both instance and feature selection.

**High-dimensional proximity pitfalls and metric learning.**   Distance concentration and hubness undermine Euclidean nearest neighbors in high-dimensional, mixed-type tables (Beyer et al., 1999; **?**; Radovanović et al., 2010). Decades of supervised metric learning show that task-aligned similarities reduce $k$-NN error by aligning neighborhoods with decision boundaries (Goldberger et al., 2005; Weinberger & Saul, 2009; Xing et al., 2002). TAAR inherits these benefits by leveraging attention as a supervised (label-aware) similarity, rather than relying on raw $X$-space proximity.

**Scaling issues: context length, label capacity, and robustness.**   Long and heterogeneous prompts can dilute useful evidence and induce interference in tabular ICL, motivating compact, information-dense contexts (Thomas et al., 2024a; Van Breugel & Van Der Schaar, 2024). Many tabular FMs also have a fixed class capacity; contemporary adaptations often require divide-and-conquer or multi-pass schemes (Ye et al., 2024).

## B   FORMAL THEOREMS AND PROOFS

### B.1   THEORETICAL PROOF OF THEOREM 2.1

We first give the proof of Theorem 2.1, in which we need the assumptions below:

(A1) The support of $S$ is compact, denoted as $\mathcal{S} \subset \mathbb{R}^m$. It has density $p_S$ with constants $0 < p_{\min} \leq p_{\max} < \infty$ such that
$$p_{\min} \leq p_S(s) \leq p_{\max} \quad \text{for } s \in \mathcal{S} \text{ a.e.}$$

(A2) Hölder smoothness ($\beta \in (0, 1]$): There exists $L > 0$ such that
$$\|\eta^\star(s) - \eta^\star(s')\|_\infty \leq L\|s - s'\|_2^\beta, \qquad \forall s, s' \in \mathcal{S}.$$
Here $\eta^\star(s) := (\eta_1^\star(s), \ldots, \eta_K^\star(s)) \in \Delta^{K-1}$ is the Bayes posterior.

(A3) Tsybakov margin condition: Let $\Delta(s) = \eta_{(1)}^\star(s) - \eta_{(2)}^\star(s) \in [0, 1]$, where $\eta_{(1)}^\star \geq \eta_{(2)}^\star$ are the top two components. There exists $C_{\mathrm{mar}} > 0$ and a noise exponent $\alpha \geq 0$ such that
$$\mathsf{P}\big(\Delta(S) \leq t\big) \leq C_{\mathrm{mar}} t^\alpha, \qquad \forall t > 0.$$
Moreover, we also assume that for any measurable random threshold $T = T(S) \in [0, 1]$, we have
$$\mathsf{P}\big(\Delta(S) \leq T \mid T\big) \leq C_{\mathrm{mar}} T^\alpha.$$

*Remark* B.1. Assumption (A1) ensures a lower and upper bound on the density to control sample size in local neighborhoods; (A2) governs the local smoothness of the Bayes posterior; Assumption (A3) simply requires that the probability mass near the Bayes boundary is not too large; intuitively, most samples are not arbitrarily close to the decision boundary.

Below are some lemmas used in proving Theorem 2.1.

**Lemma B.1** (k-Nearest Neighbor Radius). *Define the k-th nearest neighbor radius $R_k(s) = \min\{r > 0 : |\{i : \|S_i - s\| \leq r\}| \geq k\}$. Under Assumption (A1), for any $s \in \mathcal{S}$ and $k \in \{1, \ldots, n\}$, let $r_0 = \left(\frac{2k}{np_{\min}v_m}\right)^{1/m}$, where $v_m$ is the volume of the unit ball in $\mathbb{R}^m$. Then, for all $\delta \in (0, 1)$, there exists a constant $c > 0$ such that*
$$\mathsf{P}\big(R_k(s) > r_0\big) \leq \exp(-ck).$$

*Proof.* Let $N(s, r) := \sum_{i=1}^n \mathbf{1}\{\|S_i - s\| \leq r\} \sim \mathrm{Bin}\big(n, \mu(B(s, r))\big)$, where $B(s, r) := \{x \in \mathbb{R}^d : \|x - s\| \leq r\}$ and $\mu$ is the distribution of $S$. By Assumption (A1), $\mu(B(s, r)) \geq p_{\min}\mathrm{Vol}(B(s, r)) = p_{\min}v_m r^m$. For $r = r_0$,
$$\mathbb{E}[N(s, r_0)] \geq np_{\min}v_m r_0^m = 2k.$$
Using the Chernoff lower tail bound, picking $c = \frac{1}{4}$ we have:
$$\mathsf{P}\big(N(s, r_0) \leq k\big) \leq \mathsf{P}\big(N(s, r_0) \leq \frac{1}{2}\mathbb{E}[N(s, r_0)]\big) \leq \exp(-ck).$$
Since $R_k(s) > r_0 \iff N(s, r_0) < k$, the result follows. $\qquad\square$

**Lemma B.2** (k-NN Regression Bias). *Under Assumption (A2), if the event $\{R_k(s) \leq r\}$ holds, then*
$$\big\|\mathbb{E}\big[\widehat{\eta}_k(s) \mid \{S_i\}\big] - \eta^\star(s)\big\|_\infty \leq Lr^\beta.$$

*Proof.* Conditioning on $\{S_i\}$, the neighbor set $\mathcal{N}_k(s)$ is fixed. By Assumption (A2) we have:
$$\left\|\frac{1}{k}\sum_{i \in \mathcal{N}_k(s)} \eta^\star(S_i) - \eta^\star(s)\right\|_\infty \leq \frac{1}{k}\sum_{i \in \mathcal{N}_k(s)} L\|S_i - s\|_2^\beta \leq Lr^\beta.$$
Since $\mathbb{E}[\mathbf{e}(Y_i) \mid S_i] = \eta^\star(S_i)$, the result follows. $\qquad\square$

**Lemma B.3** (k-NN Regression Variance). *Conditioned on $\{S_i\}$ and $\mathcal{N}_k(s)$, $\{\mathbf{e}(Y_i)\}_{i \in \mathcal{N}_k(s)}$ are independent and take values in $[0, 1]^K$. For any $\delta \in (0, 1)$ and coordinate $c \in [K]$,*
$$\mathsf{P}\left(\big|[\widehat{\eta}_k(s)]_c - \mathbb{E}\big([\widehat{\eta}_k(s)]_c \mid \{S_i\}\big)\big| \geq t \mid \{S_i\}\right) \leq 2\exp(-2kt^2).$$
*Thus, with probability larger than $1 - \delta$, we have*
$$\big\|\widehat{\eta}_k(s) - \mathbb{E}\big(\widehat{\eta}_k(s) \mid \{S_i\}\big)\big\|_\infty \leq \sqrt{\frac{\log(2K/\delta)}{2k}}.$$

*Proof.* Conditioning on $\{S_i\}$, $[\widehat{\eta}_k(s)]_c$ is the mean of $k$ variables in $[0,1]$. The result follows from Hoeffding's inequality and applying union bound. $\qquad\square$

**Lemma B.4** (Regression Error to Classification Error). *Let $\widehat{\eta} : \mathcal{S} \to \Delta^{K-1}$ be measurable, and define the plug-in classifier $\hat{f} = \arg\max_k \widehat{\eta}_k(\cdot)$. Under Assumption (A3), there exists a constant $C > 0$ such that*

$$\mathcal{R}(\hat{f}) - \mathcal{R}(f^\star) \leq C \cdot \mathbb{E}\left[\|\widehat{\eta}(S) - \eta^\star(S)\|_\infty^{1+\alpha}\right].$$

*Here $\mathcal{R}(f) := \mathsf{P}(f(S) \neq Y)$ is defined as the classification error rate.*

*Proof.* Since $\mathcal{R}(\hat{f}|s) = 1 - \eta^\star_{\hat{f}(s)}(s), \mathcal{R}(f^\star|s) = 1 - \eta^\star_{(1)}(s)$, taking expectation on both sides we have

$$\mathcal{R}(\hat{f}) - \mathcal{R}(f^\star) = \mathbb{E}\big[\eta^\star_{(1)}(S) - \eta^\star_{\hat{f}(S)}(S)\big]$$

Define the Bayes optimal class $i^\star(s) = \arg\max_k \eta^\star_k(s)$ and $j(s) := \arg\max_k \widehat{\eta}_k(s)$. Define the regression error $\delta(s) = \|\widehat{\eta}(s) - \eta^\star(s)\|_\infty \in [0,1]$, then we have

$$\eta^\star_{i^\star} - \eta^\star_j = (\eta^\star_{i^\star} - \widehat{\eta}_{i^\star}) + (\widehat{\eta}_{i^\star} - \widehat{\eta}_j) + (\widehat{\eta}_j - \eta^\star_j) \leq \delta + 0 + \delta = 2\delta,$$

therefore we have $\eta^\star_{(1)}(s) - \eta^\star_{\hat{f}(s)}(s) \leq 2\delta(s)$ for any $s$.

Denote the event $E(s) = \{f(s) \neq f^\star(s)\}$. When $\hat{f} \neq f^\star$ we have $j \neq i^\star$, hence $\Delta = \eta^\star_{(1)} - \eta^\star_{(2)} \leq \eta^\star_{(1)} - \eta^\star_j \leq 2\delta$. Thus we have event $E(s)$ implies $\Delta(s) \leq 2\delta(s)$; When $\hat{f} = f^\star$ we have $\eta^\star_{(1)} - \eta^\star_{\hat{f}} = 0$, so we have $\eta^\star_{(1)} - \eta^\star_{\hat{f}} = (\eta^\star_{(1)} - \eta^\star_{\hat{f}})\mathbf{1}\{E\}$ for any $s$. Combining with the above result, we have

$$\eta^\star_{(1)} - \eta^\star_{\hat{f}} = (\eta^\star_{(1)} - \eta^\star_{\hat{f}})\mathbf{1}\{E\} \leq 2\delta\,\mathbf{1}\{\Delta \leq 2\delta\},$$

Taking expectation on both sides, we have

$$\mathcal{R}(\hat{f}) - \mathcal{R}(f^\star) = \mathbb{E}\big[\eta^\star_{(1)} - \eta^\star_{\hat{f}}\big] = \mathbb{E}\big[(\eta^\star_{(1)} - \eta^\star_{\hat{f}})\mathbf{1}\{E\}\big] \leq \mathbb{E}\big[2\delta(S)\,\mathbf{1}\{\Delta(S) \leq 2\delta(S)\}\big].$$

Using tower property and Assumption (A3), we have

$$\mathbb{E}\big[2\delta\,\mathbf{1}\{\Delta \leq 2\delta\}\big] = \mathbb{E}\big[2\delta\,\mathsf{P}(\Delta \leq 2\delta \mid \delta)\big] \leq \mathbb{E}\big[2\delta \cdot C_{\mathrm{mar}}(2\delta)^\alpha\big] = 2^{1+\alpha}C_{\mathrm{mar}}\,\mathbb{E}\big[\delta^{1+\alpha}\big]$$

That is, we have

$$\mathcal{R}(\hat{f}) - \mathcal{R}(f^\star) \leq C \cdot \mathbb{E}\left[\|\widehat{\eta}(S) - \eta^\star(S)\|_\infty^{1+\alpha}\right].$$

by picking $C = 2^{1+\alpha}C_{\mathrm{mar}}$. $\qquad\square$

We formally state the Theorem 2.1 as below:

**Theorem B.5** (Formal version of Theorem 2.1). *Under Assumption (A1)–(A3), if $k = k_n \to \infty$ and $k_n/n \to 0$, then we have*

$$\mathcal{R}(\hat{f}_{k\text{-}NN}) - \mathcal{R}(f^\star) \xrightarrow[n\to\infty]{} 0.$$

*Furthermore, choosing*

$$k_n \asymp n^{\frac{2\beta}{2\beta+m}},$$

*there exists a constant $C > 0$ (depending on $L, p_{\min}, p_{\max}, \beta, m, K$) such that*

$$\mathcal{R}(\hat{f}_{k\text{-}NN}) - \mathcal{R}(f^\star) \leq C \cdot n^{-\frac{\beta(1+\alpha)}{2\beta+m}}.$$

*Proof.* For an independent test point $S$, by Lemma A.1 with $r = (2k/(np_{\min}v_m))^{1/m}$ we have $\mathsf{P}(R_k(S) > r) \le e^{-ck}$. Thus we have

$$\mathbb{E}\big[R_k(S)^\beta\big] \le r^\beta + \mathrm{diam}(\mathcal{S})^\beta e^{-ck} \lesssim \left(\frac{k}{n}\right)^{\beta/m}.$$

Since $\mathbb{E}\big[\widehat{\eta}_k(S) \mid \{S_i\}\big] = \frac{1}{k}\sum_{i\in\mathcal{N}_k(S)}\eta^\star(S_i)$, we have the decomposition

$$\big\|\widehat{\eta}_k(S) - \eta^\star(S)\big\|_\infty \le \Big\|\tfrac{1}{k}\sum_{i\in\mathcal{N}_k(S)}\eta^\star(S_i) - \eta^\star(S)\Big\|_\infty + \big\|\widehat{\eta}_k(S) - \mathbb{E}[\widehat{\eta}_k(S)\mid\{S_i\}]\big\|_\infty.$$

When the event $\{R_k(S) \le r\}$ holds, by Lemma A.2 we have

$$\Big\|\tfrac{1}{k}\sum_{i\in\mathcal{N}_k(S)}\eta^\star(S_i) - \eta^\star(S)\Big\|_\infty \le L\cdot r^\beta \lesssim \left(\frac{k}{n}\right)^{\beta/m}.$$

By Lemma A.3 we have

$$\mathsf{P}\left(\big|[\widehat{\eta}_k(S)] - \mathbb{E}\big([\widehat{\eta}_k(S)]\mid\{S_i\}\big)\big| \ge t \mid \{S_i\}\right) \le 2K\exp(-2kt^2).$$

So we have $\mathbb{E}\Big[\big\|\widehat{\eta}_k(S) - \mathbb{E}(\widehat{\eta}_k(S)\mid\{S_i\})\big\|_\infty \Big| S_i\Big] \lesssim \sqrt{\frac{\log(2K)}{k}}$. and

$$\mathbb{E}\big\|\widehat{\eta}_k(S) - \eta^\star(S)\big\|_\infty \lesssim \left(\frac{k}{n}\right)^{\beta/m} + k^{-1/2}.$$

Using Lemma A.4 and choosing $k \asymp n^{\frac{2\beta}{2\beta+m}}$, we have:

$$\mathcal{R}(\hat{f}_{k\text{-NN}}) - \mathcal{R}(f^\star) \le C\cdot\mathbb{E}\big[\|\widehat{\eta}_k - \eta^\star\|_\infty^{1+\alpha}\big] \lesssim C\cdot\left(\left(\frac{k}{n}\right)^{\beta/m} + k^{-1/2}\right)^{1+\alpha} \asymp C\cdot n^{-\frac{\beta(1+\alpha)}{2\beta+m}}.$$

$\square$

For Corollary 2.2, the corollary follows simply by replacing dimension $m$ with $d$ in all above analysis.

### B.2 THEORETICAL PROOF OF THEOREM 2.3

For the kernel regression part, for notation simplicity we denote the feature vector as $X = (S, Z) \in \mathbb{R}^m \times \mathbb{R}^q \cong \mathbb{R}^d$. Denote $\eta^\star(s) = (\eta_1^\star(s), \ldots, \eta_K^\star(s)) \in \Delta^{K-1}$, then we have $\mathbb{E}\big[\mathbf{e}(Y_i) \mid S_i\big] = \eta^\star(S_i)$. We also made some fair assumptions below:

- (B1) **(i.i.d.)**: The input data $(X_i, Y_i)$s are i.i.d.
- (B2) **(positive and continuous density)**: The joint density of $X$ exists at $x_{\text{test}}$ with $f_X(x_{\text{test}}) > 0$ and is continuous in a neighborhood of $x_{\text{test}}$; the marginal density of $S$ at $s_{\text{test}}$ has $f_S(s_{\text{test}}) > 0$ and is continuous. (No requirement for $S$ and $Z$ to be independent.)
- (B3) **(regression function smoothness)**: For each $c$, $\eta_c^\star(\cdot)$ is Hölder-$\beta$ $(0 < \beta \le 2)$ in a neighborhood of $s_{\text{test}}$, i.e.,
  $$|\eta_c^\star(s) - \eta_c^\star(s')| \le L\,\|s - s'\|^\beta.$$
  In vector form, $\|\eta^\star(s) - \eta^\star(s')\|_2 \le \sqrt{K}\,L\,\|s - s'\|^\beta$.

(B4) **(kernel regularity)**: $\kappa$ is bounded, monotonically decreasing, Lipschitz; denote $\kappa(0) = K_{\max} < \infty$.

(B5) **(embedding isometry)**: The map $\phi : \mathcal{S} \to \mathbb{R}^m$ is a bi-Lipschitz, that is, there exist constants $0 < a \le b < \infty$, for $s, s' \in U$,

$$a\|s - s'\| \ \le \ \|\phi(s) - \phi(s')\| \ \le \ b\|s - s'\|,$$

and the kernel scalar function $\kappa : [0, \infty) \to (0, \infty)$ is a radial, monotonically decreasing, Lipschitz, bounded kernel scalar function

(B6) **(k scaling)**: As $n \to \infty$, $k = k(n) \to \infty$, and $k/n \to 0$.

To prove Theorem 2.3, we first prove three lemmas below:

**Lemma B.6.** *Define the "k-th Neighbor Radius" in S-space and X-space as*

$$R_S^{(k)}(s_{test}) \ := \ \min\left\{r : \ \#\{i \le n : \ \|\phi(S_i) - \phi(s_{test})\| \le r\} \ \ge \ k\right\}.$$

$$R_X^{(k)}(x_{test}) \ := \ \min\left\{r : \ \#\{i \le n : \ \|X_i - x_{test}\| \le r\} \ \ge \ k\right\}.$$

*respectively. Then under Assumptions (B1)-(B6) as $n \to \infty$, $k \to \infty$, $k/n \to 0$, we have*

$$R_S^{(k)}(s_{test}) \ = \ \Theta_p\!\left(\left(\frac{k}{n}\right)^{1/m}\right), \quad R_X^{(k)}(x_{test}) \ = \ \Theta_p\!\left(\left(\frac{k}{n}\right)^{1/d}\right).$$

*In particular, when $q \ge 1$, $R_X^{(k)}(x_{test}) \gg R_S^{(k)}(s_{test})$.*

*Proof.* By Assumption (B2) and Lebesgue differentiation theorem, there exists $\delta > 0$ such that for $r \in (0, \delta)$ we have

$$\Pr(\|S - s_{\text{test}}\| \le r) \ = \ f_S(s_{\text{test}}) \, V_m \, r^m \, \big(1 + o(1)\big),$$
$$\Pr(\|X - x_{\text{test}}\| \le r) \ = \ f_X(x_{\text{test}}) \, V_d \, r^d \, \big(1 + o(1)\big)$$

where $V_d$ is the volume of the unit ball in $\mathbb{R}^d$; $o(1)$ is with respect to $r \to 0$.

Let $N_S(r) := \sum_{i=1}^n \mathbf{1}\{\|\phi(S_i) - \phi(s_{\text{test}})\| \le r\}$, then under Assumption (B1) we have $N_S(r) \sim \text{Bin}\big(n, P_S(r)\big)$, where $P_S(r) := \Pr(\|\phi(S) - \phi(s_{\text{test}})\| \le r)$.

By Assumption (B5), for small $r$ we have

$$c_{S,-} \, r^m \ \le \ P_S(r) \ \le \ c_{S,+} \, r^m, \quad c_{S,\pm} := f_S(s_{\text{test}}) \, V_m \cdot \begin{cases} b^{-m} \, (1 - \epsilon_r), \\ a^{-m} \, (1 + \epsilon_r). \end{cases}$$

where $\epsilon_r \to 0$ as $r \to 0$.

Similarly, $N_X(r) = \sum_{i=1}^n \mathbf{1}\{\|X_i - x_{\text{test}}\| \le r\} \sim \text{Bin}\big(n, P_X(r)\big)$, with

$$c_{X,-} \, r^d \ \le \ P_X(r) \ \le \ c_{X,+} \, r^d, \quad c_{X,\pm} := f_X(x_{\text{test}}) \, V_d \, (1 \pm \epsilon_r).$$

For any $\delta \in (0, 1)$, let $r_S^-(k) := \left(\frac{(1-\delta)\,k}{n\,c_{S,+}}\right)^{1/m}, r_S^+(k) := \left(\frac{(1+\delta)\,k}{n\,c_{S,-}}\right)^{1/m}$. Then we have $\mathbb{E}[N_S(r_S^-(k))] = n \cdot P_S(r_S^-(k)) \le (1-\delta)k$ and $\mathbb{E}[N_S(r_S^+(k))] = n \cdot P_S(r_S^+(k)) \ge (1+\delta)k$. Using Chernoff bounds there exists a constant $c > 0$ such that

$$\Pr\big(N_S(r_S^-) \ge k\big) \ \le \ \exp(-c\,k), \quad \Pr\big(N_S(r_S^+) \le k\big) \ \le \ \exp(-c\,k).$$

By monotonicity ($N_S(r)$ increases with $r$, $N_S(R_S^{(k)}(s_{\text{test}})) = k$), with probability at least $1 - 2e^{-ck}$,

$$r_S^-(k) \leq R_S^{(k)}(s_{\text{test}}) \leq r_S^+(k).$$

Similarly in $X$-space, if we define $r_X^-(k) := \left(\frac{(1-\delta)\,k}{n\,c_{X,+}}\right)^{1/(d)}, r_X^+(k) := \left(\frac{(1+\delta)\,k}{n\,c_{X,-}}\right)^{1/(d)}$, then with probability at least $1 - 2e^{-ck}$ we have

$$r_X^-(k) \leq R_X^{(k)}(x_{\text{test}}) \leq r_X^+(k).$$

when $k \to \infty$ we have $1 - 2e^{-ck} \to 1$. So the result follows. $\qquad\square$

Let the truncated index set for a method (1 or 2) be $\mathcal{I}$ (size $k$), with the corresponding radius upper bound denoted as $R$. Let the weights be

$$w_i := \frac{K_S(s_{\text{test}}, S_i)}{\sum_{j\in\mathcal{I}} K_S(s_{\text{test}}, S_j)}, \quad i \in \mathcal{I},$$

so $\sum_{i\in\mathcal{I}} w_i = 1$, and the corresponding estimator is $\widehat{\eta}(s_{\text{test}}) = \sum_{i\in\mathcal{I}} w_i\,\mathbf{e}(Y_i)$.

**Lemma B.7.** *For method 2, with corresponding radius upper bounded by $R_S^{(k)}(s_{test})$, there exists constants $C_b, C_v < \infty$ such that under Assumption (B1)-(B6), as $n \to \infty$, $k \to \infty$, $k/n \to 0$, we have*

$$\mathbb{E}\big[\|\widehat{\eta}^{(2)}(s_{test}) - \eta^\star(s_{test})\|_2^2\big] \leq C_b^2\,R_S^{(k)}(s_{test})^{2\beta} + \frac{C_v}{k} + o\Big(R_S^{(k)}(s_{test})^{2\beta} + \frac{1}{k}\Big).$$

*Proof.* For simplicity we ignore the superscripts (2) below. By bias and variance decomposition, we have

$$\mathbb{E}\big[\|\widehat{\eta}(s_{\text{test}}) - \eta^\star(s_{\text{test}})\|_2^2\big] = \big\|\mathbb{E}[\widehat{\eta}(s_{\text{test}})] - \eta^\star(s_{\text{test}})\big\|_2^2 + \mathbb{E}\big[\|\widehat{\eta}(s_{\text{test}}) - \mathbb{E}\widehat{\eta}(s_{\text{test}})\|_2^2\big].$$

For simplicity we denote $R = R_S^{(k)}(s_{\text{test}})$. For $i \in \mathcal{I}$, we have $\|\phi(S_i) - \phi(s_{\text{test}})\| \leq R$, so by monotonicity of $\kappa$ we have $K_{\min}(R) := \kappa(R) \leq K_S(s_{\text{test}}, S_i) \leq \kappa(0) =: K_{\max}$. Thus we have

$$\frac{K_{\min}(R)}{k\,K_{\max}} \leq w_i \leq \frac{K_{\max}}{k\,K_{\min}(R)}, \quad \sum_{i\in\mathcal{I}} w_i^2 \leq k \cdot \frac{K_{\max}^2}{k^2 K_{\min}(R)^2} = \frac{K_{\max}^2}{k\,K_{\min}(R)^2}.$$

By Lemma A.6 we know that as $n \to \infty$ we have $R \to 0$, so $K_{\min}(R) \to K_{\max}$.

For the bias upper bound, conditioned on $\{S_i\}_{i\in\mathcal{I}}$ we have $\mathbb{E}\big[\widehat{\eta}(s_{\text{test}}) \,\big|\, \{S_i\}\big] = \sum_{i\in\mathcal{I}} w_i\,\eta^\star(S_i)$, thus we have

$$\mathbb{E}\big[\widehat{\eta}(s_{\text{test}}) - \eta^\star(s_{\text{test}}) \,\big|\, \{S_i\}\big] = \sum_{i\in\mathcal{I}} w_i\big(\eta^\star(S_i) - \eta^\star(s_{\text{test}})\big).$$

By Assumption (B3) and $\|\phi(S_i) - \phi(s_{\text{test}})\| \leq R$,

$$\|\eta^\star(S_i) - \eta^\star(s_{\text{test}})\|_2 \leq \sqrt{K}\,L\,\|S_i - s_{\text{test}}\|^\beta \leq \sqrt{K}\,L\left(\frac{R}{a}\right)^\beta.$$

Therefore we have $\left\|\mathbb{E}\big[\widehat{\eta}(s_{\text{test}}) - \eta^\star(s_{\text{test}}) \,\big|\, \{S_i\}\big]\right\|_2 \leq \sqrt{K}\,L\left(\frac{R}{a}\right)^\beta$. Denote $Z = \mathbb{E}\big[\widehat{\eta}(s_{\text{test}}) - \eta^\star(s_{\text{test}}) \,\big|\, \{S_i\}\big]$, since $\mathbb{E}[Z] = \mathbb{E}[\widehat{\eta}(s_{\text{test}})] - \eta^\star(s_{\text{test}})$, taking expectation over $\{S_i\}$, we have

$$\left\|\mathbb{E}[\widehat{\eta}(s_{\text{test}})] - \eta^\star(s_{\text{test}})\right\|_2 = \|\mathbb{E}[Z]\|_2 \leq \mathbb{E}[\|Z\|_2] \leq C_b\,R^\beta, \quad C_b := \sqrt{K}\,L\,a^{-\beta}.$$

Therefore, the square of bias is less than $C_b^2 R^{2\beta}$

For the variance upper bound, let $\xi_i := \mathbf{e}(Y_i) - \eta^\star(S_i)$, then $\mathbb{E}[\xi_i \mid S_i] = 0$ and they are conditionally independent across $i$. Since $\widehat{\eta}(s_{\text{test}}) = \sum_{i \in \mathcal{I}} w_i \, \mathbf{e}(Y_i)$ and $\mathbb{E}[\mathbf{e}(Y_i) \mid S_i] = \eta^\star(S_i)$, we have

$$\widehat{\eta}(s_{\text{test}}) - \mathbb{E}[\widehat{\eta}(s_{\text{test}}) \mid \{S_i\}] = \sum_{i \in \mathcal{I}} w_i \, \xi_i.$$

By conditionally independence, the second moment conditioned on $\{S_i\}$ is

$$\mathbb{E}\Big[\big\| \sum_{i \in \mathcal{I}} w_i \, \xi_i \big\|_2^2 \,\Big|\, \{S_i\}\Big] = \sum_{i \in \mathcal{I}} w_i^2 \, \mathbb{E}\big[\|\xi_i\|_2^2 \mid S_i\big],$$

where

$$\mathbb{E}\big[\|\xi_i\|_2^2 \mid S_i\big] = \sum_{c=1}^m \eta_c^\star(S_i)(1 - \eta_c^\star(S_i)) = \sum_{c=1}^m \eta_c^\star(S_i) - \sum_{c=1}^m \eta_c^\star(S_i)^2 = 1 - \sum_{c=1}^m \eta_c^\star(S_i)^2 \leq 1.$$

as $\mathbb{E}[(\mathbf{1}\{Y_i = c\} - \eta_c^\star(S_i))^2 \mid S_i] = \eta_c^\star(S_i)(1 - \eta_c^\star(S_i))^2 + (1 - \eta_c^\star(S_i))\eta_c^\star(S_i)^2 = \eta_c^\star(S_i)(1 - \eta_c^\star(S_i))$. Thus,

$$\mathbb{E}\Big[\big\| \sum_i w_i \, \xi_i \big\|_2^2 \,\Big|\, \{S_i\}\Big] \leq \sum_{i \in \mathcal{I}} w_i^2 \leq \frac{K_{\max}^2}{k \, K_{\min}(R)^2}.$$

Taking expectation over $\{S_i\}$, the variance term is

$$\mathbb{E}\big[\|\widehat{\eta}(s_{\text{test}}) - \mathbb{E}\widehat{\eta}(s_{\text{test}})\|_2^2\big] \leq \frac{K_{\max}^2}{k} \cdot \mathbb{E}\Big[\frac{1}{K_{\min}(R)^2}\Big].$$

Hence there exists a constant $C_v < \infty$ such that

$$\mathbb{E}\big[\|\widehat{\eta}(s_{\text{test}}) - \mathbb{E}\widehat{\eta}(s_{\text{test}})\|_2^2\big] \leq \frac{C_v}{k}.$$

$\square$

**Lemma B.8.** *For method 1, with corresponding radius upper bounded by $R_X^{(k)}(x_{test})$, there exists constants $C_b, C_v < \infty$ such that under Assumption (B1)–(B6), as $n \to \infty$, $k \to \infty$, $k/n \to 0$, we have*

$$\mathbb{E}\big[\|\widehat{\eta}^{(1)}(s_{test}) - \eta^\star(s_{test})\|_2^2\big] \leq C_b^2 \, R_X^{(k)}(x_{test})^{2\beta} + \frac{C_v}{k} + o\Big(R_X^{(k)}(x_{test})^{2\beta} + \frac{1}{k}\Big).$$

*Proof.* The proof is almost identical to Lemma B.7. As we have $a\|s - s'\| \leq \|\phi(s) - \phi(s')\| \leq b\|s - s'\|$ for $s, s' \in U$, in the bias part the constants will not affect the final order. The proof of the variance part is exactly the same. $\square$

We formally state the Theorem 2.3 as below:

**Theorem B.9.** *Under Assumption (B1)–(B6), we have*

$$\text{MSE}_1(s_{test}) := \mathbb{E}\big[\|\widehat{\eta}^{(1)}(s_{test}) - \eta^\star(s_{test})\|_2^2\big] \leq \Theta\big(n^{-\frac{2\beta}{2\beta+d}}\big),$$

$$\text{MSE}_2(s_{test}) := \mathbb{E}\big[\|\widehat{\eta}^{(2)}(s_{test}) - \eta^\star(s_{test})\|_2^2\big] \leq \Theta\big(n^{-\frac{2\beta}{2\beta+m}}\big)$$

*Proof.* Pick $k \asymp n^{\frac{d}{2\beta+d}}$ in Method 1 and $k \asymp n^{\frac{m}{2\beta+m}}$ in Method 2, by Lemma B.2 and Lemma B.3 we have

$$\mathrm{MSE}_1(s_{\text{test}}) := \mathbb{E}\big[\|\widehat{\eta}^{(1)}(s_{\text{test}}) - \eta^\star(s_{\text{test}})\|_2^2\big] \lesssim C_b''\left(\frac{n^{\frac{d}{2\beta+d}}}{n}\right)^{\frac{2\beta}{d}} + \frac{C_v}{n^{\frac{d}{2\beta+d}}} = \Theta\big(n^{-\frac{2\beta}{2\beta+d}}\big),$$

$$\mathrm{MSE}_2(s_{\text{test}}) := \mathbb{E}\big[\|\widehat{\eta}^{(2)}(s_{\text{test}}) - \eta^\star(s_{\text{test}})\|_2^2\big] \lesssim C_b'\left(\frac{n^{\frac{m}{2\beta+m}}}{n}\right)^{\frac{2\beta}{m}} + \frac{C_v}{n^{\frac{m}{2\beta+m}}} = \Theta\big(n^{-\frac{2\beta}{2\beta+m}}\big)$$

which is what we want to prove. $\square$

## C    MORE RESULTS AND DETAILS OF EXPERIMENTS

**Implementation Details**    We adopt a standardized 70:10:20 split for training, validation, and testing across all datasets. To ensure a fair comparison, all baseline models are used in their base configurations without ensembling, additional transformations or fine-tuning. For baseline methods, we perform hyperparameter optimization over 10 trials, selecting the best configuration based on validation AUC. Specifically, for the KNN-based method, we search the number of neighbors $k \in \{50, 100, 200, 500, 1000\}$. Likewise, for TAAR, we define a search space for its retrieval-related hyperparameters. The optimal setup for each method, determined by validation performance, is then evaluated on the test set.

We begin by computing the attention score for each test sample. For feature selection, we retain the top features whose cumulative attention scores account for a proportion of $\eta_{feature\_attn}$ of the total attention. Then, to determine the number of samples to retrieve per test instance, we apply two dynamic thresholds: one based on the cumulative attention scores, ensuring that the selected samples contribute at least $\eta_{sample\_attn}$ of the total attention, and the other based on the sample count, requiring that the number of retrieved samples constitutes $\eta_{cont}$ of the total available context samples. The final number of samples retrieved is taken as the maximum of the two values determined by these thresholds. Next, we group the test samples using K-means clustering, where the number of clusters is set to $\beta$, and the grouping is based on the overlap among their retrieved sample sets. Finally, inference is carried out independently for each cluster, and the results are aggregated to produce the final output. The corresponding hyperparameter search space is provided in Tab. 3.

Table 3: Hyperparameter Search Space for Retrieval Module

| Hyperparameter | Search Space |
| --- | --- |
| $\eta_{\text{feature\_attn}}$ | {0.5, 0.6, 0.7, 0.8, 0.9} |
| $\eta_{\text{sample\_attn}}$ | {0.5, 0.6, 0.7, 0.8, 0.9} |
| $\eta_{\text{cont}}$ | {0.10, 0.15, 0.20, 0.25} |
| $\beta$ | {5, 10, 20, 30, 40} |

### C.1    THE RESULT OF BCCO-CLS BENCHMARKS

In this section, we present the performance differences between TAAR methods and the baseline on the BCCO-CLS benchmark. The experimental result shown in Tab. 4 demonstrates that TAAR exhibits significant advantages on the BCCO-CLS benchmark, both with LimiX and TabPFN-v2. Additionally, we conducted a supplementary experiment to compare the performance of TAAR when LimiX and TabPFN-v2 utilizes an ensembling approach. The result shown in Tab. 5 reveals that TAAR substantially enhances model performance even under the ensembling setting.

Table 4: Performance on BCCO-CLS benchmark. Red and green values indicate performance improvement and degradation relative to the baseline method, respectively.

| Method | AUC (↑) | Acc.(↑) | f1 (↑) | AUC-Rank (↓) | Acc.-Rank (↓) | f1-Rank (↓) |
|---|---|---|---|---|---|---|
| LimiX | 0.854 | 0.781 | 0.693 | 5.377 | 4.981 | 5.443 |
| +KNN | 0.859(+0.005) | 0.789(+0.008) | 0.709(+0.016) | 4.453(-0.924) | 3.991(-0.990) | 4.387(-1.056) |
| +TAAR(ours) | **0.872** (+0.018) | **0.809**(+0.028) | **0.735**(+0.042) | **1.840**(-3.537) | **1.415**(-6.566) | **1.594**(-3.849) |
| TabPFN-v2 | 0.837 | 0.766 | 0.671 | 7.528 | 6.783 | 7.057 |
| +KNN | 0.832(-0.005) | 0.766(+0.000) | 0.666(-0.005) | 8.358(+0.830) | 7.453(+0.670) | 7.811(+0.754) |
| +TAAR(ours) | **0.848**(+0.011) | **0.777**(+0.011) | **0.681**(+0.010) | **5.330**(-2.198) | **5.755**(-1.028) | **6.245**(-0.812) |
| TabICL | 0.840 | 0.760 | 0.661 | 6.425 | 7.311 | 8.038 |
| Mitra | 0.841 | 0.769 | 0.679 | 7.019 | 7.142 | 7.519 |
| XGBoost | 0.834 | 0.762 | 0.674 | 7.538 | 7.509 | 7.500 |
| CatBoost | 0.829 | 0.757 | 0.664 | 8.717 | 8.226 | 8.123 |
| LightGBM | 0.832 | 0.763 | 0.678 | 8.019 | 7.321 | 7.368 |
| TabR | 0.804 | 0.744 | 0.652 | 10.047 | 9.217 | 8.896 |

Table 5: Performance on BCCO-CLS benchmark with ensembling method. *_ES means model utilizes an ensembling approach.

| Method | AUC (↑) | AUC-Rank (↓) |
|---|---|---|
| LimiX_ES | 0.861 | 4.632 |
| +KNN | 0.861(+0.000) | 4.538(-0.094) |
| +TAAR(ours) | **0.873**(+0.012) | **1.783**(-2.849) |
| TabPFN-v2_ES | 0.843 | 7.264 |
| +KNN | 0.839(-0.004) | 8.132(+0.868) |
| +TAAR(ours) | **0.856**(+0.013) | **4.840**(-2.424) |
| TabICL | 0.840 | 6.547 |
| Mitra | 0.841 | 7.075 |
| XGBoost | 0.834 | 7.604 |
| LightGBM | 0.832 | 8.085 |
| CatBoost | 0.829 | 8.802 |
| ModernNCA | 0.813 | 9.868 |
| TabR | 0.804 | 10.066 |

## C.2 DETAILS OF EXPERIMENT DATASETS

This section details the datasets used across all experiments and presents the corresponding results obtained with LimiX-TAAR.

### C.2.1 STANDARD BENCHMARKS

This section presents the details of the benchmarks used in main experiments, as detailed in Tabs. 6, 8 and 9.

Table 6: Details of BCCO-CLS benchmark.

| Dataset | Cont. | Feat. | Class | AUC |
|---|---|---|---|---|
| BNG(cmc) | 44236 | 9 | 3 | 0.763 |
| CPMP-2015-runtime-classification | 368 | 22 | 4 | 0.723 |
| CostaMadre1 | 207 | 37 | 2 | 0.816 |
| Fitness_Club_c | 1200 | 6 | 2 | 0.827 |
| GAMETES_Epistasis_2-Way_20atts_0.1H_EDM-1_1 | 1120 | 20 | 2 | 0.733 |
| Gender_Gap_in_Spanish_WP | 3796 | 13 | 3 | 0.709 |
| GesturePhaseSegmentationProcessed | 6911 | 32 | 5 | 0.948 |
| KDD | 4025 | 45 | 2 | 0.910 |
| LED-display-domain-7digit | 350 | 7 | 10 | 0.961 |
| MIC | 1319 | 104 | 2 | 0.895 |
| Marketing_Campaign | 1792 | 27 | 2 | 0.903 |
| National_Health_and_Nutrition_Health_Survey | 1822 | 7 | 2 | 0.778 |
| PieChart1 | 493 | 37 | 2 | 0.957 |
| PopularKids | 334 | 10 | 3 | 0.714 |
| SPECTF | 244 | 44 | 2 | 0.933 |
| ad-click-data | 700 | 3 | 2 | 0.950 |
| ad-click-prediction-dataset | 7000 | 7 | 2 | 1.000 |
| ada | 2902 | 48 | 2 | 0.925 |
| airlines_seed_0_nrows_2000_nclasses_10_ncols_100_stratify_True | 1600 | 7 | 2 | 0.684 |
| amazon-survey-coupon-recommendation-dataset | 8878 | 25 | 2 | 0.818 |
| analcatdata_apnea2 | 332 | 3 | 2 | 0.920 |
| analcatdata_birthday | 255 | 3 | 2 | 0.970 |
| analcatdata_broadwaymult | 199 | 6 | 7 | 0.874 |
| analcatdata_dmft | 462 | 4 | 5 | 0.579 |
| analcatdata_draft | 256 | 4 | 2 | 0.658 |
| analcatdata_germangss | 280 | 5 | 4 | 0.775 |
| arsenic-female-bladder | 391 | 4 | 2 | 0.850 |
| autoMpg | 278 | 7 | 2 | 0.984 |
| autoUniv-au4-2500 | 2000 | 100 | 3 | 0.749 |
| autoUniv-au7-1100 | 880 | 12 | 5 | 0.761 |
| balance-scale | 437 | 4 | 3 | 0.997 |
| bankadditionalfullcsv | 28831 | 20 | 2 | 0.952 |
| baseball | 1072 | 16 | 3 | 0.918 |
| chemical-x | 1231 | 10 | 2 | 0.905 |
| chscase_funds | 129 | 2 | 2 | 0.704 |
| chscase_vine2 | 327 | 2 | 2 | 0.922 |
| classification-in-asteroseismology | 700 | 1 | 2 | 0.977 |
| colic | 257 | 26 | 2 | 0.979 |
| colleges_usnews | 911 | 32 | 2 | 0.822 |
| companion-plants | 696 | 3 | 4 | 1.000 |
| company_bankruptcy_prediction | 5455 | 95 | 2 | 0.967 |
| compass | 13315 | 17 | 2 | 0.923 |
| consultant-job-placement-data-2023 | 307 | 8 | 2 | 0.814 |
| coupons | 105 | 5 | 3 | 0.954 |
| credit-approval | 483 | 15 | 2 | 0.924 |
| customer-segmentation | 8068 | 9 | 4 | 0.579 |
| cylinder-bands | 378 | 35 | 2 | 0.950 |
| dis | 3017 | 29 | 2 | 0.995 |
| disclosure_x_noise | 463 | 3 | 2 | 0.556 |

| Dataset | Cont. | Feat. | Class | AUC |
|---|---|---|---|---|
| employees-attrition-analysis | 3087 | 27 | 2 | 0.999 |
| eucalyptus | 515 | 19 | 5 | 0.942 |
| eye_movements_bin | 6086 | 20 | 2 | 0.916 |
| fictional-character-battle-outcome-prediction | 1645 | 7 | 2 | 0.856 |
| first-order-theorem-proving | 4282 | 51 | 6 | 0.828 |
| forty_soybean_cultivars_from_subsequent_harvests | 224 | 10 | 2 | 1.000 |
| glioma_grading_clinical_and_mutation_features_dataset | 587 | 23 | 2 | 0.948 |
| golf_play_dataset_extended | 876 | 9 | 2 | 0.984 |
| guitar-chord-finger-positioning | 205 | 9 | 7 | 1.000 |
| hcc-survival-data-set | 115 | 30 | 2 | 0.735 |
| heart-h | 205 | 12 | 2 | 0.883 |
| heart-long-beach | 140 | 13 | 5 | 0.715 |
| hill-valley | 969 | 100 | 2 | 0.940 |
| income | 22384 | 12 | 2 | 0.928 |
| jasmine | 2088 | 144 | 2 | 0.887 |
| jm1 | 8708 | 21 | 2 | 0.753 |
| liver-disorders | 408 | 10 | 2 | 0.800 |
| loan-application-data | 350 | 11 | 2 | 0.813 |
| machine-learning | 14000 | 12 | 2 | 0.718 |
| madeline | 2198 | 259 | 2 | 0.970 |
| malware-analysis-datasets-pe-section-headers | 30305 | 4 | 2 | 0.973 |
| maternal-health-risk | 709 | 6 | 3 | 0.967 |
| medium-app-reviews-from-google-play-store | 42886 | 7 | 4 | 0.787 |
| meta | 369 | 21 | 2 | 0.970 |
| mfeat-morphological | 1600 | 6 | 10 | 0.974 |
| mfeat-zernike | 1600 | 47 | 10 | 0.988 |
| microaggregation2 | 16000 | 20 | 5 | 0.809 |
| nfl-combine-performance-data-2009-2019 | 2433 | 15 | 2 | 0.797 |
| orange-quality | 168 | 10 | 8 | 0.850 |
| pbc | 292 | 18 | 2 | 0.856 |
| pbcseq | 1361 | 17 | 2 | 0.955 |
| pc1 | 887 | 21 | 2 | 0.921 |
| phoneme | 3782 | 5 | 2 | 0.970 |
| plasma_retinol | 220 | 13 | 2 | 0.606 |
| pm10 | 350 | 7 | 2 | 0.715 |
| predict-customer-purchase-behavior-dataset | 1050 | 6 | 2 | 0.881 |
| predict_students_dropout_and_academic_success | 3539 | 34 | 3 | 0.832 |
| prnn_synth | 175 | 2 | 2 | 0.956 |
| profb | 470 | 9 | 2 | 0.765 |
| qsar-bioconcentration-classes-data-set | 545 | 11 | 3 | 0.910 |
| retail-chain-salespeople-engagement | 1466 | 64 | 2 | 0.658 |
| rmftsa_ctoarrivals | 184 | 2 | 2 | 0.998 |
| schizo | 238 | 13 | 2 | 1.000 |
| sentiment-analysis | 349 | 5 | 3 | 0.833 |
| servo | 116 | 4 | 2 | 0.989 |
| simple-diabetes-dataset | 537 | 8 | 2 | 0.815 |
| sonar | 145 | 60 | 2 | 0.976 |
| spambase | 3680 | 57 | 2 | 0.991 |
| statlog | 800 | 20 | 2 | 0.801 |
| tabletpc-priceclassification | 1400 | 19 | 4 | 0.997 |
| tokyo1 | 671 | 44 | 2 | 0.983 |
| turiye_student_evaluation | 4656 | 32 | 5 | 0.746 |
| user-knowledge | 282 | 5 | 5 | 1.000 |

| Dataset | Cont. | Feat. | Class | AUC |
|---|---|---|---|---|
| waveform-5000 | 4000 | 40 | 3 | 0.977 |
| wholesale-customers | 308 | 8 | 2 | 0.975 |
| wine-quality-dataset-balanced-classification | 14700 | 11 | 7 | 0.899 |
| wine-quality-red | 1279 | 4 | 6 | 0.818 |

Table 8: Details of Tabzilla benchmarks

| Dataset | Cont. | Feat. | Class | AUC |
|---|---|---|---|---|
| Australian | 621 | 14 | 2 | 0.964 |
| Bioresponse | 3375 | 1776 | 2 | 0.900 |
| GesturePhaseSegmentationProcessed | 8885 | 32 | 5 | 0.964 |
| SpeedDating | 7540 | 120 | 2 | 0.890 |
| ada_agnostic | 4105 | 48 | 2 | 0.916 |
| artificial-characters | 9196 | 7 | 10 | 0.999 |
| balance-scale | 562 | 4 | 3 | 1.000 |
| cnae-9 | 972 | 856 | 9 | 0.999 |
| colic | 331 | 26 | 2 | 0.947 |
| credit-approval | 621 | 15 | 2 | 0.948 |
| credit-g | 900 | 20 | 2 | 0.867 |
| electricity | 40780 | 8 | 2 | 0.988 |
| elevators | 14939 | 18 | 2 | 0.944 |
| heart-h | 264 | 13 | 2 | 0.871 |
| jasmine | 2685 | 144 | 2 | 0.885 |
| jungle_chess_2pcs_raw_endgame_complete | 40337 | 6 | 3 | 0.974 |
| kc1 | 1898 | 21 | 2 | 0.893 |
| mfeat-fourier | 1800 | 76 | 10 | 0.992 |
| mfeat-zernike | 1800 | 47 | 10 | 0.986 |
| monks-problems-2 | 540 | 6 | 2 | 1.000 |
| nomao | 31018 | 118 | 2 | 0.997 |
| phoneme | 4863 | 5 | 2 | 0.985 |
| profb | 604 | 9 | 2 | 0.839 |
| qsar-biodeg | 949 | 41 | 2 | 0.960 |
| socmob | 1040 | 5 | 2 | 0.997 |
| splice | 2871 | 60 | 3 | 0.997 |
| vehicle | 761 | 18 | 4 | 0.982 |

Table 9: Details of Talent benchmarks

| Dataset | Cont. | Feat. | $R^2$ |
|---|---|---|---|
| 1000-Cameras-Dataset | 830 | 10 | 0.701 |
| 2dplanes | 32614 | 10 | 0.945 |
| 3D_Estimation_using_RSSI_of_WLAN_dataset | 4608 | 6 | 0.949 |
| 3D_Estimation_using_RSSI_of_WLAN_dataset_complete_1_target | 11520 | 12 | 0.891 |
| Abalone_reg | 3341 | 8 | 0.591 |

| Dataset | Cont. | Feat. | $R^2$ |
|---|---|---|---|
| Ailerons | 11000 | 40 | 0.858 |
| Another-Dataset-on-used-Fiat-500-(1538-rows) | 1230 | 6 | 0.866 |
| BNG(echoMonths) | 13996 | 8 | 0.480 |
| BNG(lowbwt) | 24883 | 9 | 0.608 |
| BNG(stock) | 47239 | 9 | 0.807 |
| Bias_correction_r | 6180 | 21 | 0.968 |
| Bias_correction_r_2 | 6180 | 21 | 0.966 |
| Brazilian_houses_reproduced | 8553 | 8 | 0.999 |
| CPMP-2015-regression | 1686 | 25 | 0.912 |
| CPS1988 | 22524 | 6 | 0.306 |
| CookbookReviews | 14545 | 7 | 0.054 |
| Data_Science_Salaries | 3004 | 5 | 0.155 |
| Diamonds | 43152 | 9 | 0.983 |
| Facebook_Comment_Volume | 32759 | 53 | 0.753 |
| Food_Delivery_Time | 36474 | 8 | 0.344 |
| Goodreads-Computer-Books | 987 | 5 | 0.367 |
| IEEE80211aa-GATS | 3236 | 27 | 0.994 |
| Job_Profitability | 11584 | 28 | 0.289 |
| Kaggle_bike_sharing_demand_challange | 8708 | 9 | 0.883 |
| Laptop_Prices_Dataset | 3552 | 8 | 0.773 |
| Large-scale_Wave_Energy_Farm_Perth_100 | 5821 | 201 | 0.990 |
| Large-scale_Wave_Energy_Farm_Sydney_100 | 1854 | 201 | 0.984 |
| Large-scale_Wave_Energy_Farm_Sydney_49 | 14371 | 99 | 0.995 |
| MIP-2016-regression | 872 | 144 | 0.435 |
| MiamiHousing2016 | 11145 | 16 | 0.949 |
| Mobile_Phone_Market_in_Ghana | 2880 | 14 | 0.902 |
| Moneyball | 985 | 14 | 0.952 |
| NASA_PHM2008 | 36734 | 21 | 0.652 |
| NHANES_age_prediction | 1821 | 7 | 0.389 |
| OnlineNewsPopularity | 31715 | 59 | 0.032 |
| Parkinson_Multiple_Sound_Recording | 832 | 26 | 0.242 |
| Parkinsons_Telemonitoring | 4700 | 19 | 0.999 |
| Physicochemical_r | 36584 | 9 | 0.756 |
| SAT11-HAND-runtime-regression | 3552 | 116 | 0.815 |
| Shop_Customer_Data | 1600 | 6 | -0.001 |
| Superconductivty | 16957 | 81 | 0.925 |
| Wine_Quality_red | 1279 | 11 | 0.405 |
| Wine_Quality_white | 3918 | 11 | 0.580 |
| airfoil_self_noise | 1202 | 5 | 0.972 |
| analcatdata_supreme | 3241 | 7 | 0.973 |
| archive2 | 914 | 12 | 0.654 |
| archive_r56_Portuguese | 520 | 30 | 0.349 |
| auction_verification | 1634 | 7 | 0.995 |
| avocado_sales | 14599 | 13 | 0.954 |
| bank32nh | 6553 | 32 | 0.584 |
| bank8FM | 6553 | 8 | 0.965 |
| boston | 404 | 13 | 0.870 |
| chscase_foot | 420 | 5 | 0.006 |
| colleges | 5650 | 44 | 0.592 |
| combined_cycle_power_plant | 7654 | 4 | 0.972 |
| communities_and_crime | 1595 | 102 | 0.683 |
| concrete_compressive_strength | 824 | 8 | 0.963 |
| cpu_act | 6553 | 21 | 0.984 |

| Dataset | Cont. | Feat. | $R^2$ |
|---|---|---|---|
| cpu_small | 6553 | 12 | 0.978 |
| dataset_sales | 8590 | 10 | 0.620 |
| debutanizer | 1915 | 7 | 0.973 |
| delta_elevators | 7613 | 6 | 0.659 |
| elevators | 13279 | 18 | 0.915 |
| fifa | 14450 | 5 | 0.677 |
| fried | 32614 | 10 | 0.956 |
| garments_worker_productivity | 957 | 13 | 0.549 |
| gas_turbine_CO_and_NOx_emission | 29386 | 10 | 0.999 |
| healthcare_insurance_expenses | 1070 | 6 | 0.877 |
| house_16H_reg | 18227 | 16 | 0.725 |
| house_8L | 18227 | 8 | 0.741 |
| house_prices_nominal | 1168 | 79 | 0.899 |
| house_sales_reduced | 17290 | 18 | 0.914 |
| houses | 16512 | 8 | 0.876 |
| housing_price_prediction | 436 | 12 | 0.632 |
| kin8nm | 6553 | 8 | 0.893 |
| mauna-loa-atmospheric-co2 | 1780 | 6 | 0.998 |
| mv | 32614 | 10 | 0.999 |
| pol_reg | 12000 | 48 | 0.991 |
| pole | 11998 | 26 | 0.991 |
| puma32H | 6553 | 32 | 0.946 |
| puma8NH | 6553 | 8 | 0.661 |
| qsar_aquatic_toxicity | 436 | 8 | 0.677 |
| qsar_fish_toxicity | 726 | 6 | 0.571 |
| satellite_image | 5148 | 36 | 0.927 |
| sensory | 460 | 11 | 0.372 |
| socmob | 924 | 5 | 0.945 |
| space_ga | 2485 | 6 | 0.795 |
| steel_industry_energy_consumption | 28032 | 10 | 0.999 |
| stock | 760 | 9 | 0.993 |
| stock_fardamento02 | 5021 | 6 | 0.552 |
| sulfur | 8064 | 6 | 0.902 |
| topo_2_1 | 7108 | 266 | 0.046 |
| treasury | 839 | 15 | 0.996 |
| us_crime | 1595 | 126 | 0.705 |
| volume | 40794 | 53 | 0.619 |
| weather_izmir | 1168 | 9 | 0.993 |
| wind | 5259 | 14 | 0.825 |
| wine+quality | 5197 | 11 | 0.518 |
| yprop_4_1 | 7108 | 251 | 0.061 |

### C.2.2 LARGE-LABEL-SPACE DATASETS

This section presents the details of the benchmarks used in large-label-space datasets experiments, as detailed in Tab. 11.

Table 11: large-label-space sample datasets details

| dataset name | context | feature | classes | Acc. |
|---|---|---|---|---|
| one-hundred-plants-shape | 1280 | 64 | 100 | 0.941 |
| texture | 4400 | 40 | 11 | 1.000 |
| kr-vs-k | 22444 | 6 | 18 | 0.867 |
| letter | 16000 | 15 | 26 | 0.981 |
| one-hundred-plants-margin | 1280 | 64 | 100 | 0.983 |
| internet_usage | 8086 | 70 | 46 | 0.745 |
| helena | 52156 | 27 | 100 | 0.596 |
| kropt | 22444 | 6 | 18 | 0.860 |
| ASP-POTASSCO-classification | 1035 | 141 | 11 | 0.821 |
| UJI_Pen_Characters | 1091 | 80 | 35 | 0.860 |
| one-hundred-plants-texture | 1279 | 64 | 100 | 0.970 |
| walking-activity | 119465 | 4 | 22 | 0.837 |

## C.3 MASSIVE SAMPLE SIZE DATASETS EXPERIMENT

This section provides a detailed introduction to the massive sample size datasets experiment, including the datasets used and the experimental results. Additionally, we conduct validation experiments on medium-scale datasets to demonstrate the effectiveness of our method.

We present the details of the datasets used in large-label-space experiments in Tab. 12 and the results obtained with LimiX-TAAR.

Table 12: massive sample datasets details

| index | dataset name | context | feature | classes | AUC |
|---|---|---|---|---|---|
| 1 | airlines-dataset-to-predict-a-delay | 539383 | 7 | 2 | 0.741 |
| 2 | covid19-dataset | 1048575 | 20 | 7 | 0.710 |
| 3 | employee-attrition-dataset | 74498 | 22 | 2 | 0.856 |
| 4 | kickstarter-projects | 374853 | 7 | 5 | 0.883 |
| 5 | loantap-data | 396030 | 24 | 2 | 0.735 |
| 6 | roadside-noise-level-dataset-with-labels | 417000 | 1 | 2 | 0.689 |
| 7 | stellar-classification-dataset-sdss17 | 100000 | 8 | 3 | 0.829 |
| 8 | titanic-huge-dataset-1m-passengers | 1000000 | 8 | 2 | 0.909 |
| 9 | smoking-drinking-dataset | 991346 | 23 | 3 | 0.898 |
| 10 | expedia-travel-dataset | 100000 | 22 | 2 | 0.735 |

The result shown in Fig. 6 demonstrate that integrating TAAR consistently improves performance over the baseline LimiX model, with further gains observed in several datasets when using KNN-based augmentation.

### C.3.1 VALIDATION EXPERIMENTS

In this section, we conduct validation experiments on both real-world and synthetic data. We perform retrieval using a divide-and-conquer strategy as well as a full-sample retrieval on the entire dataset. The retrieval process for both methods are carried out on a per-test-sample basis.

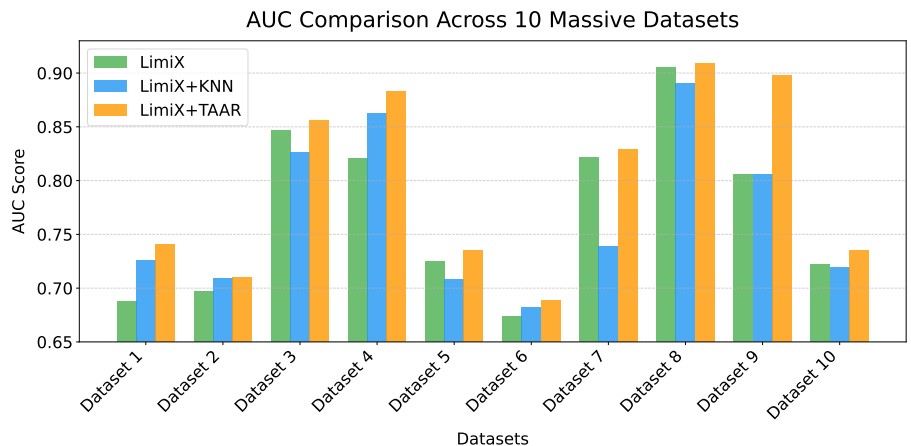

Figure 6: Comparison of AUC scores across 10 massive datasets.

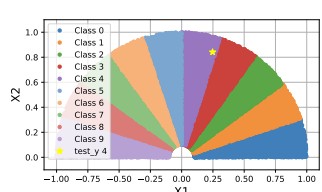

(a) Visualization of synthetic samples. Categories are color coded by sector. The test sample is marked by a yellow pentagram.

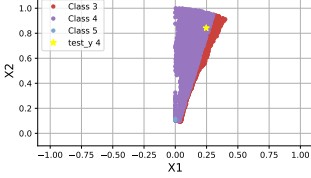

(b) Top 10% of in-context samples sorted by TAAR-method.

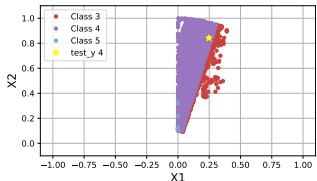

(c) Top 10% of in-context samples sorted by divide-and-conquer strategy.

Figure 7: Visualization of retrieval based on different method.

**The real-world dataset validation.** We select a dataset from BCCO-CLS containing over 20,000 context samples for testing. The average overlap rate of the selected samples reached 96.3%. The AUC scores for the two methods reached 0.972 and 0.971, respectively, significantly outperforming the baseline of 0.959. This demonstrates that our strategy achieves near-lossless retrieval performance while successfully adhering to the GPU memory constraints.

**Synthetic data analysis.** Following the approach in Sec. 3.1, we perform retrieval using the two methods and plot the results from both on the same figure. We observe a high degree of overlap between the samples retrieved by the two methods. The visualization of the experiment is shown in Sec. C.3.1

## C.4 RUNTIME STUDY

We analyze the computational overhead in our main experiments. Theoretically, the total number of forward passes is linear in the number of clusters $n$, $O(n + 1)$, leading to a proportional relationship between cluster count and inference time. Empirically, we measure the total wall-clock time on the entire benchmark using an NVIDIA H20 GPU and compare our method against KNN and baseline. The result is shown as Tab. 13

Table 13: Runtime of different method on Tabzilla classification benchmark per test sample

| Method | Inference time(ms/it) |
|---|---|
| LimiX | 6.31 |
| +KNN | 356.97 |
| +TAAR(ours) | 71.30 |

## C.5 ANALYSIS OF DIFFERENT LAYER ATTENTION

In this section, we employ the attention mechanisms from layers 1 to 12 to perform TAAR. As shown in the Fig. 8, the results indicate that using the attention from the last layer yields the best performance.

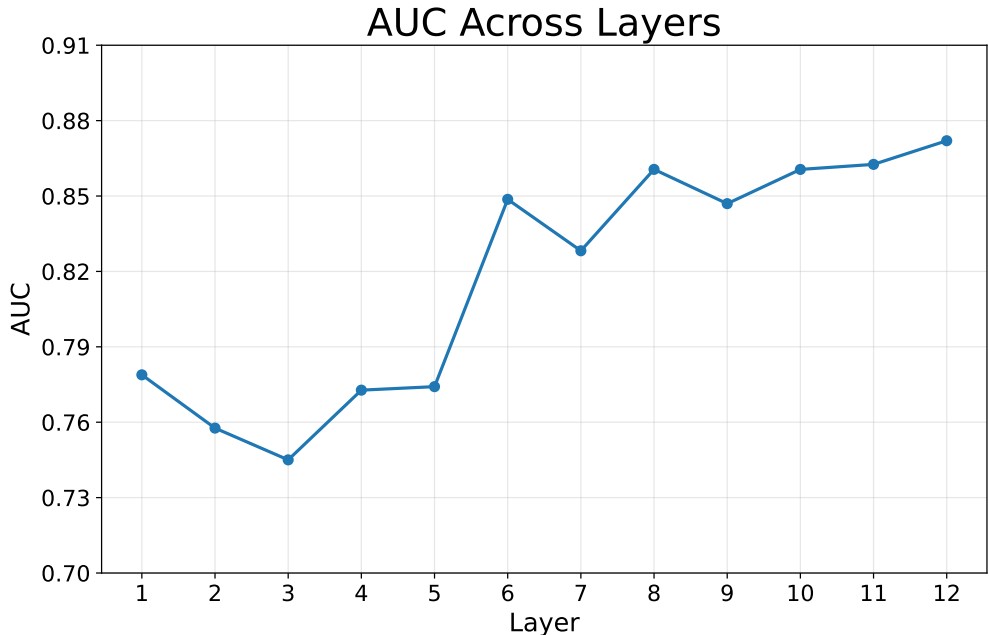

Figure 8: The AUC values across layers 1 to 12 of the LimiX model on the BCCO-CLS benchmark.

