# OpenReview forum: "Task-Aligned Attention Retrieval for Scaling Tabular Foundation Models"
_ICLR.cc/2026/Conference — Submitted to ICLR 2026_

### Official Review · Reviewer_Dkxm · 2025-10-31

**Soundness:** 3
**Presentation:** 3
**Contribution:** 3
**Rating:** 6
**Confidence:** 4

**Summary:**

This paper proposes Task-Aligned Attention Retrieval (TAAR), a training-free retrieval module for tabular foundation models (e.g., TabPFN-v2, LimiX). Instead of picking neighbors by Euclidean distance in input space, TAAR uses the model’s own attention to (i) select a query-specific feature subspace and (ii) retrieve the most task-relevant instances. A companion scheme, Class-Range Lifting Retrieval (CRLR), uses a regression head to build a label-sparse local context so classification backbones with small fixed vocabularies (≤10 classes) can handle many-class problems at inference. The method needs only an extra forward pass, includes a streaming/stratified variant for large pools, and comes with theory showing better rates vs. distance-only neighbors.

**Strengths:**

1. The paper is original in shifting tabular in-context retrieval from geometry-based neighbors to task-aligned, attention-guided retrieval, and in combining feature-level and instance-level attention into a single retrieval procedure.

2. It is training-free : The proposed TAAR can be plugged into existing tabular FMs (TabPFN-v2, LimiX) without re-training, which raises the practical value of the contribution.

3. The paper removes a real limitation of many tabular FMs—small fixed label vocabularies—via CRLR, making >10-class problems feasible at inference.

4. The empirical section is broad: across TabZilla, BCCO-CLS and TALENT benchmarks, task-aligned retrieval consistently improves LiMix and TabPFNv2.

**Weaknesses:**

1. Computational overhead. The method requires running a retrieval procedure for every test sample, which can be expensive in realistic serving scenarios (low-latency or high-throughput settings).

2. Strong reliance on backbone attention quality. TAAR assumes that the backbone’s attention already encodes task-relevant feature and instance importance. On datasets with noisy features, domain shift, or weakly trained tabular FMs, attention may be poorly calibrated, and the task-aligned retrieval could become unstable or even harmful.

3. CRLR presumes the regression head can reliably propose a label-sparse local context. On long-tailed, highly discrete, or noisy-label datasets this assumption may not hold.

**Questions:**

(1) LimiX and TabPFNv2 share broadly similar architectures, yet TAAR brings a much larger improvement to LimiX. Can the authors clarify whether this is because LimiX inherently produces more informative / better attention maps (i.e., it is already better at identifying salient features and relevant training instances) than TabPFNv2 ?

(2) Why KNN helps LimiX but often hurts TabPFN-v2 ? In Tables 1 and 2, KNN-based retrieval consistently improves LimiX, but for TabPFN-v2 it often degrades performance.

(3) Suppose we run TAAR on top of LimiX and obtain, for each test sample, a set of task-relevant training instances. If we then feed exactly these retrieved instances to other tabular FMs (e.g., TabPFN-v2, TabICL), do we also observe gains? In other words, is the importance signal that LimiX exposes via attention at least partly model-agnostic, or is TAAR’s benefit mostly tied to the backbone that generated the attention?

(4) I like the overall idea, but I still have concerns about practicality: running retrieval for every test sample can be expensive in real deployments. Do the authors have concrete ideas for reducing the cost of TAAR ?

(5) For LimiX and TabPFNv2, which layer’s attention map is actually used to drive TAAR ?

(6) The citation for TabICL is currently incorrect. The right one should be:

Jingang, Q. U., Holzmüller, D., Varoquaux, G., & Le Morvan, M. TabICL: A Tabular Foundation Model for In-Context Learning on Large Data. In Forty-second International Conference on Machine Learning.

---

> ### Author Response · Authors · 2025-11-21
> **Response to Reviewer Dkxm**
>
> Thank you for your thoughtful review and constructive suggestions. We have revised the manuscript accordingly and have uploaded the updated version. In response to the specific points raised, we provide the following point-by-point clarifications.
>
> Weakness 1:
>
> Computational overhead.
>
> Response 1:
>
> We employed a clustering-based approach to significantly reduce computational time. The core idea is to enable multiple test samples to share the same set of context samples. As detailed in Appendix C, our method computes the attention scores for all samples in a single forward pass. These attention scores are then used for clustering, whereby samples within the same cluster share an identical set of context samples and can be processed simultaneously during inference (see Lines 873-875). This design results in substantially lower time overhead compared to the KNN method. We have also provided a theoretical analysis of the algorithm's time complexity in Appendix C.4 to further substantiate this advantage(see Lines 1266). Through experimental comparisons between per-sample processing and our method, we demonstrate that this clustering strategy reduces inference time by 90% while maintaining comparable performance.
>
> Weakness 2:
>
> Strong reliance on backbone attention quality. On datasets with noisy features, domain shift, or weakly trained tabular FMs, attention may be poorly calibrated, and the task-aligned retrieval could become unstable or even harmful.
>
> Response 2:
>
> We acknowledge that our current work does not address strong distribution shifts, and we have clearly identified the investigation of such scenarios as an important direction for future research. Regarding the dependency on different Foundation Models (FMs), it is true that our method's performance is correlated with the quality of their inherent attention mechanisms. Our empirical results consistently show that stronger FMs generally lead to more significant performance improvements, which aligns with our methodological expectations.
>
> Weakness 3:
>
> CRLR presumes the regression head can reliably propose a label-sparse local context. On long-tailed, highly discrete, or noisy-label datasets this assumption may not hold.
>
> Response 3:
>
> We have conducted experiments to evaluate our method's performance on long-tail distributions, and the results demonstrate that it still achieves significant performance improvements. For data with highly discrete features, TAAR applies reordering and normalization during the encoding process, which effectively handles such cases without any adverse impact. Regarding the noisy label problem, similar to strong distribution shifts, this aspect is currently beyond the scope of the present study.
>
> Response to Question 1:
>
> We would like to clarify that the attention maps are computed separately for LimiX and TabPFNv2 using their own forward passes. The performance gains observed when applying TAAR to TabPFNv2 are therefore attributable to our method itself, rather than being an effect inherited from LimiX’s power.
>
> Response to Question 2:
>
> We believe this discrepancy arises from differences in model robustness to feature noise. As supported by our experiments, KNN is more susceptible to noise in the feature space compared to TAAR. It appears that LimiX has a built-in robustness that partially compensates for the noise introduced by KNN, which may explain why KNN still benefits LimiX but often degrades the performance of TabPFNv2.
>
> Response to Question 3:
>
> TAAR is designed to work with the attention mechanism of a specific backbone model. Since TabICL and some other tabular FMs do not explicitly produce feature-level attention maps, it is difficult to directly apply TAAR and realize its full potential.
>
> Response to Question 4:
>
> Please refer to Response 1.
>
> Response to Question 5:
>
> In our experiments, we use the attention maps from the final layer of both LimiX and TabPFNv2. This choice is supported by empirical evidence and related works, which indicate that the last layer’s attention best captures high-level, task-relevant patterns in tabular data.
>
> To further substantiate this, we have provided additional experiments in Appendix C.5 which empirically demonstrate the superior performance of the final attention layer. Please refer to Lines 1278 for details.
>
> Response to Question 6:
>
> We sincerely thank you for pointing out the incorrect citation. We have updated the manuscript accordingly in the latest version.

---

> > ### Comment · Reviewer_Dkxm · 2025-11-24
> >
> > Thanks for authors' response and additional experiments. You have cleared up most of my doubts, and I will raise my score accordingly.
> >
> > One more comment about the grouping / clustering of test samples to reduce the computational cost :
> >
> > This kind of grouping / clustering is very important. In the revised version, you should provide full details about it in the main paper instead of just mentioning it in the appendix.

---

### Official Review · Reviewer_QNjt · 2025-11-01

**Soundness:** 2
**Presentation:** 2
**Contribution:** 2
**Rating:** 2
**Confidence:** 3

**Summary:**

The paper introduces Task-aligned attention retrieval (TAAR), which involves subselecting samples and / or features based on the attention weights in a foundational model. Compared to using the foundational models TabPFN and Limix on all the data (without ensembling), the authors show slightly increased performance for KNN-based retrieval and even slightly better performance for TAAR.

**Strengths:**

The paper addresses an important issue for tabular data: scaling foundational models to larger sample sizes. This is one of the key open questions for the current generation of tabular models. The paper evaluates the proposed methods using two state-of-the-art foundational models for tabular data, TabPFNV2 and Limix. The authors use three standard benchmarks, TabZilla and BCCO-CLS for classification, and Talent for regression.

**Weaknesses:**

# Main concerns

- The method that is described in this paper is already present in the Limix paper. This is somewhat confusing as the authors claim it to be novel. It is not thoroughly evaluated, but it is mentioned, and Figures 1 and 2 are nearly identical to the Limix paper.

- The comparison against foundational models is not using the settings recommended by the authors of the foundational models. TabPFNV2 and Limix should both be used with the default ensembling for comparison. Disabling the ensembling means to not be representative of the actual proposed method.

- There are several strong, but unsupported statements in the paper, in particular in the introduction. For example "Retrieval is crucial for scaling foundational models". The TabFlex model for example is able to scale without retrieval. There are also several fine-tuning based approaches like tunetables and TabPFN unleashed that are able to work without retrieval.

- Using retrieval based on attention weights does not really solve the scaling problem, as the attention weights need to be computed first. The paper describes a method around this, by splitting the training data into batches. However, this weakens the motivation for retrieval.

## Minor notes
- The comparison of methods across datasets ideally would be done with a critical difference diagram.

- Feature selection is seen as an end in itself. However, the literature has shown that feature selection is usually not beneficial for ML Benchmark datasets.

- For Table 1, showing improvements as red and degradation as green seems quite confusing to me.

- The theoretical analysis for nearest neighbors retrieval seems unhelpful. Theorem 2.1 seems to show the consistency of KNN, which has a proof that can be found in the textbook "A probabilistic theory of pattern recognition L Devroye, L Györfi, G Lugosi".

- Line 144 uses but doesn't define e(Y_i) and \delta^{K-1}

- Line 308 mentions "Max cell to 5,000,000" which is unclear to me.

- Line 402 is missing a whitespace in "benchmarksto"

- Hollman 2025a and 2025b refer to the same paper and should be consolidated.

**Questions:**

- What is different from this work to the attention-based selection proposed in the Limix paper?

- What are the models used for TAAR and ECOC in Figure 4?

- Are you using the multi-label strategy implemented in TabPFNV2 for Figure 4 or are you reimplementing ECOC? How does it compare against a one-vs-rest reduction?

- Can you provide a comparison of the retrieval methods to the foundational models with ensembling? Is retrieval on top of ensembling beneficial?

**Details Of Ethics Concerns:**

The method described in this paper is part of the Limix paper. In particular, Figures 1 and 2 are basically identical to that paper. The limix paper is cited, which makes this a bit strange, and I assume this was submitted by the limix authors. If this was submitted by the limix authors, I would probably not flag it as an ethical issue, though submitting part of the paper and also citing the paper is somewhat strange.

---

> ### Author Response · Authors · 2025-11-21
> **Response to Reviewer QNjt (part1)**
>
> Thank you for your thoughtful review and constructive suggestions. We have revised the manuscript accordingly and have uploaded the updated version. In response to the specific points raised, we provide the following point-by-point clarifications.
>
> Weakness 1:
>
> The distinction between our work and the attention-based selection method mentioned in the LimiX paper.
>
> Response 1:
>
> In the LimiX paper, the approach is only roughly mentioned without providing theoretical justification or experimental validation of its effectiveness. The implementation details of the method is not thoroughly explained. In contrast, this paper offers a comprehensive description of the algorithm along with a theoretical proof of the method’s validity and feasibility. Additionally, the approach in this paper achieves improved inference speed and has been extensively validated across diverse experimental settings.
>
> Weakness 2:
>
> The comparison against foundational models is not using the settings recommended by the authors of the foundational models. TabPFNV2 and Limix should both be used with the default ensembling for comparison. Disabling the ensembling means to not be representative of the actual proposed method.
>
> Response 2:
>
> Our method is primarily compared against other retrieval-based approaches, which did not consider the ensembling techniques and used non-ensembling evaluation protocols. For a fair comparison, we report results without ensembling in the paper. To investigate whether our method can also significantly enhance the base model’s performance under an ensemble setting, we conduct validation experiments and report reuslts in BCCO-CLS benchmarks. as shown in the following table. The results show that TAAR substantially enhances model performance even under the ensembling setting. We also include the results and analysis in the manuscript in Lines 885-924.
>
> | model            | auc   | auc_rank |
> |----------------|------ |----------  |
> | LimiX_ES        | 0.861 | 4.632    |
> | +KNN             | 0.861 | 4.538    |
> | +TAAR(ours)      | 0.873 | 1.783    |
> | TabPFN-v2_ES     | 0.843 | 7.264    |
> | +KNN             | 0.839 | 8.132    |
> | +TAAR(ours)      | 0.856 | 4.840    |
> | TabICL           | 0.840 | 6.547    |
> | Mitra            | 0.841 | 7.075    |
> | XGBoost          | 0.834 | 7.604    |
> | LightGBM         | 0.832 | 8.085    |
> | CatBoost         | 0.829 | 8.802    |
> | ModernNCA        | 0.813 | 9.868    |
> | TabR             | 0.804 | 10.066   |
>
> Weakness 3:
>
> There are several strong, but unsupported statements in the paper, in particular in the introduction. For example "Retrieval is crucial for scaling foundational models". The TabFlex model for example is able to scale without retrieval. There are also several fine-tuning based approaches like tunetables and TabPFN unleashed that are able to work without retrieval.
>
> Response 3:
>
> Thank you for pointing out this issue. We believe that retrieval is an effective approach to address scaling up across three dimensions: sample, feature, and class number. Accordingly, we have revised the relevant description to "Retrieval serves as an effective approach to address various scaling challenges in in-context learning with tabular foundation models.", as shown in Lines 011. Additionally, we have revised other irrelevant description to "Empirically, TAAR achieves favorable scaling properties across feature space, sample size, and target-class cardinality. It generally outperforms state-of-the-art baselines, achieving a competitive balance between accuracy and efficiency.", as shown in Lines 071-073.

---

> > ### Author Response · Authors · 2025-11-21
> > **Response to Reviewer QNjt (part2)**
> >
> > Weakness 4:
> >
> > Using retrieval based on attention weights does not really solve the scaling problem, as the attention weights need to be computed first. The paper describes a method around this, by splitting the training data into batches. However, this weakens the motivation for retrieval.
> >
> > Response 4:
> >
> > Thank you for raising this point. We adopted a divide-and-conquer strategy to address the computational constraints: the dataset is divided into several batches, retrieval is performed separately on each batch, and the results are subsequently merged. It is worth noting that we concatenate the retrieval results from each processed batch as context for the subsequent batch. Please see Lines 335-340 for implementation details.
> >
> > The results presented in Appendix C.3 demonstrate that this approach effectively resolves the Out-Of-Memory (OOM) issue while achieving a significant improvement in AUC on these datasets compared to random sampling and KNN methods. To further validate our batch-wise strategy, we conducted additional experiments to compare both the sample selection and final performance against a full-dataset retrieval. Through experiments on medium-scale real-world datasets, we observed that the samples retrieved using this batch-wise strategy are nearly identical to those retrieved from the entire dataset at once, yet it reduces GPU memory usage by approximately 80%. This conclusion is supported by our sample-by-sample comparison of retrieval results between the two strategies for each test instance. For example, on a BCCO-CLS dataset containing over 20,000 context samples, the average overlap rate of the selected samples reached 96.3%. The AUC scores for the two methods reached 0.972 and 0.971, respectively, significantly outperforming the baseline of 0.959. This demonstrates that our strategy achieves near-lossless retrieval performance while successfully adhering to the GPU memory constraints.
> >
> > We have included this additional experiment in Appendix C.3.1.
> >
> > Question 1:
> >
> > What is different from this work to the attention-based selection proposed in the Limix paper?
> >
> > Answer 1:
> >
> > Please refer to Response 1.
> >
> > Question 2:
> >
> > What are the models used for TAAR and ECOC in Figure 4?
> >
> > Answer 2:
> >
> > Thank you for your comment. The model used is LimiX. We have now included a description of the model in the main text, please refer to Lines 353-354.
> >
> > Question 3:
> >
> > Are you using the multi-label strategy implemented in TabPFNV2 for Figure 4 or are you reimplementing ECOC? How does it compare against a one-vs-rest reduction?
> >
> > Answer 3:
> >
> > Regarding the models used for TAAR and ECOC in Figure 4, and the implementation of the multi-label strategy:
> > For multi-class tasks, both TAAR and ECOC methods are based on LimiX, Specifically, the ECOC method was implemented using the tabpfn-extension library [1] , with the model replaced by LimX. We note that there are limited OVR (one-vs-rest) algorithms available for TFMs, and ECOC is the most widely adopted among them. Therefore, we focused our comparison on the ECOC approach.
> >
> > [1] https://github.com/PriorLabs/tabpfn-extensions/tree/main/src/tabpfn_extensions/many_class
> >
> > Question 4:
> >
> > Can you provide a comparison of the retrieval methods to the foundational models with ensembling? Is retrieval on top of ensembling beneficial?
> >
> > Answer 4:
> >
> > Please refer to Response 2.
> >
> > Regarding the Minor Notes:
> >
> > We thank the reviewer for pointing out these valuable details. We have addressed all of them in our revised manuscript.

---

> ### Author Response · Authors · 2025-11-25
> **Kindly Requesting Your Review on Our Response**
>
> Thank you again for your insightful and constructive comments on our paper. Your feedback was very helpful, and we have done our best to address it in our response.
>
> We posted a reply to your comments on Nov.20 but we haven't heard back yet. We truly value your perspective and were hoping to engage in a further discussion on weakness and questions of our work to ensure we have understood and addressed your points thoroughly.
>
> We would be very grateful if you could review our response and share any further thoughts you might have. Your guidance is crucial for helping us improve this work.
>
> Thank you for your valuable time and contribution.

---

> > ### Comment · Area_Chair_fiuV · 2025-11-28
> > **Reminder to Respond to Rebuttal**
> >
> > Reviewer QNjt,
> >
> > Please ensure that you read the authors' response to your review and post an acknowledgement **before end of Dec. 2**. This date is the cutoff for reviewer posts.
> >
> > Thank you for supporting quality peer review at ICLR.
> >
> > AC

---

### Official Review · Reviewer_K55e · 2025-11-02

**Soundness:** 2
**Presentation:** 2
**Contribution:** 2
**Rating:** 2
**Confidence:** 4

**Summary:**

The paper proposes a method to deal with large feature and context size during inference for tabular foundation models (TFMs). Authors propose to use model's attention to select top feature column and/or context instances. The motivation is that attention similarity better captures information relevant to prediction than KNN is raw space. Theoretical analysis and empirical results on real-world data sets are provided to support this conclusion.

**Strengths:**

The paper proposes a simple approach to select feature and instances that are both speed inference and improve prediction accuracy for in-context TMFs. Theoretical justification is given although it is based multiple assumptions and might not hold in practice. Empirical results on real-world data show that leveraging attention is more effective than current methods of using KNN in the raw feature space.

**Weaknesses:**

I found the paper notation heavy and difficult to read. The theoretical results have multiple assumptions that likely won't hold in real world settings. The experimental section needs a significant revision. It would be useful to see an apples to apples comparison of KNN vs the proposed context selection approach (Equation 4) without feature selection or other artefacts of TAAR. That would clearly show if attention based retrieval is better than the feature one. I couldn't find any results on run time, that would also be useful to provide especially for large datasets. 3.2.2 does not refer to any figure and some figures like Figure 3 have very vague descriptions. I think a full revision is required.

**Questions:**

Do you have a direct comparison of KNN vs attention-based (Equation 4) retrieval as well as runtime overhead for TAAR?

---

> ### Author Response · Authors · 2025-11-21
> **Response to Reviewer K55e**
>
> Thank you for your thoughtful review and constructive suggestions. We have revised the manuscript accordingly and have uploaded the updated version. In response to the specific points raised, we provide the following point-by-point clarifications.
>
> Weakness1:
>
> The theoretical results have multiple assumptions that likely won't hold in real world settings.
>
> Response1:
>
> We carefully checked all the assumptions involved in the paper, and find them mild and commonly adopted in the related literature.
>
> For assumption A1, This originates from the Assumption 1 of [1], where the author states that it's necessary, since the convergence rate will be slower if the pdf can approach zero arbitrarily close; Assumption A2 and A3 both originates from paper [2], A2 (together with assumption B3 afterwards) controls how "complex" the Bayes decision boundary is, the smoother it is (larger $\beta$), the faster this nonparametric methods can learn; For assumption A3, if the true data distribution has a massive pile of points right around the decision boundary (e.g., two classes completely overlap), then indeed no method whatsoever can learn it fast. It's a mild and common assumption that is likely hold in real world settings.
>
> For assumptions in Theorem 2.3, B1, B2, B3 and B6 are clear, B4 is only a general assumption on $\kappa$, it can be Gaussian or Epanechnikov kernel and so on, B5 is crucial in our analysis. For TAAR, $\phi$ can be interpreted as a learned task-aligned representation or an attention-induced subspace. Assuming that $\phi$ does not distort the task-related geometry of $\mathcal{S}$ too drastically, this is also a reasonable modeling approach in practice.
>
> [1]. Analysis of KNN Density Estimation. Puning Zhao, Lifeng Lai. IEEE Transactions on Information Theory
>
> [2]. Fast learning rates for plug-in classifiers. Jean-Yves Audibert, Alexandre B. Tsybakov. The Annals of Statistics
>
>
> Weakness2:
>
> It would be useful to see an apples to apples comparison of KNN vs the proposed context selection approach (Equation 4) without feature selection or other artefacts of TAAR.
>
> Response2:
>
> We are afraid that this is a misunderstanding. Our method aims to achieve scale-up in both feature and sample space simultaneously. Therefore, TAAR employs a dual-attention design, transitioning from a feature-attention module to an instance-attention module (see Lines 063–067). Thus, feature-level retrieval is also an integral part of TAAR, which is not available in other methods (see Lines 096–100). Additionally, in the ablation study, we presented a variant of TAAR without the feature-retrieval module and demonstrate that it still maintains a significant advantage over KNN. We had included those results in the ablation study (Figure 5), where “NF” denotes the results using only Equation (4). It can be observed that this configuration improves the baseline by 1.2% and outperforms KNN by 0.6% (Table 1).
>
> Additionally, we noticed that “NS” (which uses feature selection alone) only brings a 0.2% improvement over the baseline, indicating that feature selection alone offers limited gains, while TAAR-NF (using only Equation (4)) yields more significant improvement. The full TAAR model achieves the best performance overall.
>
>
> Weakness3:
>
> I couldn't find any results on run time.
>
> Response3:
>
> The runtime comparison and analysis are in Appendix C.4. Please refer to Lines 1263-1275 and Table 13.
>
>
> Weakness4:
>
> 3.2.2 does not refer to any figure and some figures like Figure 3 have very vague descriptions.
>
> Response4:
>
> Thank you for pointing out the missing reference to the experimental results in Section 3.2.2 and vague description. We have updated the manuscript accordingly, (see Lines 341-342 and Lines 376-397).

---

> ### Author Response · Authors · 2025-11-25
> **Kindly Requesting Your Review on Our Response**
>
> Thank you again for your insightful and constructive comments on our paper. Your feedback was very helpful, and we have done our best to address it in our response.
>
> We posted a reply to your comments on Nov.20 but we haven't heard back yet. We truly value your perspective and were hoping to engage in a further discussion on weakness and questions of our work to ensure we have understood and addressed your points thoroughly.
>
> We would be very grateful if you could review our response and share any further thoughts you might have. Your guidance is crucial for helping us improve this work.
>
> Thank you for your valuable time and contribution.

---

> > ### Comment · Area_Chair_fiuV · 2025-11-28
> > **Reminder to Respond to Rebuttal**
> >
> > Reviewer K55e,
> >
> > Please ensure that you read the authors' response to your review and post an acknowledgement **before end of Dec. 2**. This date is the cutoff for reviewer posts.
> >
> > Thank you for supporting quality peer review at ICLR.
> >
> > AC

---

### Meta-Review · Area_Chair_4qmE · 2025-12-23

**Summary:**

This paper proposes Task-Aligned Attention Retrieval (TAAR) to scale tabular foundation models by selecting task-relevant features and context instances using model attention, along with an extension (CRLR) for many-class classification. Reviewers agreed that the problem of scaling tabular foundation models is important and that the idea of leveraging attention for retrieval is intuitive. However, the decision was informed by several substantial concerns: (i) limited and unclear novelty, particularly due to strong overlap with prior work, (ii) questionable experimental protocol choices, including non-standard evaluation settings and incomplete or unclear baselines, (iii) theoretical and practical limitations, especially around assumptions, computational overhead, and reliance on well-calibrated attention, and (iv) presentation and clarity issues, including heavy notation, vague figure descriptions, and key implementation details relegated to the appendix. While one reviewer viewed the contribution more favorably after rebuttal, the overall consensus was that the paper does not yet meet the bar for a clear, well-substantiated contribution at ICLR.

**Reviewer Concerns:**

The rebuttal addressed several specific technical questions, including providing ablation results isolating instance-level attention retrieval, adding runtime analysis and batch-wise/clustered inference strategies, clarifying implementation details, revising overly strong claims in the introduction, and correcting missing descriptions, figures, and citations. These responses helped clarify the method and partially mitigated concerns about computational overhead and evaluation under ensembling.

However, key concerns remain outstanding. Multiple reviewers were unconvinced that TAAR is sufficiently novel relative to prior work, especially LimiX, with similarities in both conceptual approach and presentation. The experimental protocol remains contentious, as results without default ensembling and limited “apples-to-apples” comparisons weaken the empirical claims. The theoretical analysis was viewed by some reviewers as relying on assumptions with limited practical relevance, and its added value over standard KNN theory was questioned. Finally, the method’s dependence on high-quality attention, scalability in real-world deployment, and generality beyond the tested backbones remain open concerns. Taken together, these unresolved issues outweigh the addressed points.

**Reviewer Scores:**

Reviewer K55e raised major concerns about theory realism, experimental clarity, and missing runtime comparisons. While the rebuttal addressed several factual issues, the reviewer’s core reservations about practicality and presentation would likely remain.
Reviewer QNjt identified overlap with prior work, questioned novelty and evaluation choices, and raised potential research integrity concerns. Although the rebuttal clarified differences and added ensemble results, it is unlikely these would change their overall assessment.
Reviewer Dkxm initially rated the paper marginally above threshold and, after rebuttal and additional experiments, explicitly indicated they would raise their score. This reviewer would likely move to a clear accept-level score, contingent on better integration of key details into the main paper.
Overall, despite one reviewer becoming more positive after rebuttal, the majority of reviewers would not substantially change their scores, and the balance of opinion remains below the acceptance threshold.

---

### Decision · Program_Chairs · 2026-01-26

Reject